# Metabolic-Dysfunction-Associated Steatotic Liver Disease: Molecular Mechanisms, Clinical Implications, and Emerging Therapeutic Strategies

**DOI:** 10.3390/ijms26072959

**Published:** 2025-03-25

**Authors:** Jeysson E. Mejía-Guzmán, Ramón A. Belmont-Hernández, Norberto C. Chávez-Tapia, Misael Uribe, Natalia Nuño-Lámbarri

**Affiliations:** 1Translational Research Unit, Medica Sur Clinic & Foundation, Mexico City 14050, Mexico; jey18112000@gmail.com (J.E.M.-G.); belher_2094ra@hotmail.com (R.A.B.-H.); nchavezt@medicasur.org.mx (N.C.C.-T.); 2Postgraduate Program in Experimental Biology, División de Ciencias Básicas y de la Salud (DCBS), Universidad Autonoma Metropolitana-Iztapalapa, Mexico City 09340, Mexico; 3Obesity and Digestive Diseases Unit, Medica Sur Clinic & Foundation, Mexico City 14050, Mexico; muribe@medicasur.org.mx; 4Surgery Department, Faculty of Medicine, The National Autonomous University of Mexico (UNAM), Mexico City 04510, Mexico

**Keywords:** MASLD, lipogenesis, chronic inflammation, oxidative stress, biomarkers, emerging therapies

## Abstract

Metabolic-dysfunction-associated steatotic liver disease (MASLD), previously known as non-alcoholic fatty liver disease (NAFLD), is a highly prevalent metabolic disorder characterized by hepatic steatosis in conjunction with at least one cardiometabolic risk factor, such as obesity, type 2 diabetes, hypertension, or dyslipidemia. As global rates of obesity and metabolic syndrome continue to rise, MASLD is becoming a major public health concern, with projections indicating a substantial increase in prevalence over the coming decades. The disease spectrum ranges from simple steatosis to metabolic-dysfunction-associated steatohepatitis (MASH), fibrosis, cirrhosis, and hepatocellular carcinoma, contributing to significant morbidity and mortality worldwide. This review delves into the molecular mechanisms driving MASLD pathogenesis, including dysregulation of lipid metabolism, chronic inflammation, oxidative stress, mitochondrial dysfunction, and gut microbiota alterations. Recent advances in research have highlighted the role of genetic and epigenetic factors in disease progression, as well as novel therapeutic targets such as peroxisome proliferator-activated receptors (PPARs), fibroblast growth factors, and thyroid hormone receptor beta agonists. Given the multifaceted nature of MASLD, a multidisciplinary approach integrating early diagnosis, molecular insights, lifestyle interventions, and personalized therapies is critical. This review underscores the urgent need for continued research into innovative treatment strategies and precision medicine approaches to halt MASLD progression and improve patient outcomes.

## 1. Introduction

Recently, the condition previously known as non-alcoholic fatty liver disease (NAFLD) was renamed metabolic-dysfunction-associated steatotic liver disease (MASLD). This change was driven by the need to address several limitations of the original terminology and to better represent the disease and its associated factors. Previously, the definition of NAFLD excluded other liver diseases, such as hepatitis B or C, as well as conditions caused by factors like alcoholism or certain medications. Furthermore, the terms “non-alcoholic” and “fatty” were deemed stigmatizing for some patients. Most importantly, the strict exclusion of alcohol consumption above specific thresholds overlooked patients with a moderate alcohol intake who still met the criteria for the disease [1,2,3].

To overcome these limitations, experts employed the Delphi method to establish a new nomenclature and definition that is both inclusive and representative. This process was spearheaded by three major organizations: the American Association for the Study of Liver Diseases (AASLD), the European Association for the Study of the Liver (EASL), and the Asociación Latinoamericana para el Estudio del Hígado (ALEH). In collaboration with regulatory agencies, patient advocacy groups, and multidisciplinary medical experts, these organizations reached a consensus. As a result, the term “steatotic liver disease” was adopted as a general designation for all causes of hepatic steatosis, and MASLD officially replaced NAFLD [1,4,5].

In recent years, a paradigm shift has redefined the classification of fatty liver diseases, grouping them under the broader term of steatotic liver disease. This reclassification includes alcohol-associated liver disease (ALD), MASLD, and a newly recognized condition called metabolic-dysfunction-and-alcohol-related liver disease (MetALD).

ALD refers to liver injury caused by chronic excessive alcohol consumption, with thresholds established by the World Health Organization and major liver societies. ALD is typically diagnosed when alcohol intake exceeds 21 standard drinks per week for men and 14 drinks per week for women. MetALD, on the other hand, is a newly introduced entity that identifies individuals with both metabolic dysfunction, such as obesity, type 2 diabetes, hypertension, or dyslipidemia, and moderate alcohol consumption, defined as intake levels below the ALD thresholds but above complete abstinence. This classification reflects growing evidence that metabolic risk factors, even in the presence of moderate alcohol intake, contribute synergistically to liver injury. By formally recognizing this overlap, MetALD provides a clearer diagnostic framework for a patient population that previously lacked precise classification within the traditional categories of alcohol-related or non-alcoholic liver disease [6,7].

The new MASLD definition is based on the presence of hepatic steatosis along with at least one cardiometabolic risk factor. These include obesity, type 2 diabetes (T2D), hypertension, hypertriglyceridemia, or low HDL cholesterol. Obesity is defined as a body mass index (BMI) of ≥25 kg/m^2^, or ≥23 kg/m^2^ in Asian populations, or a waist circumference exceeding 94 cm in men and 80 cm in women, with ethnicity-specific adjustments. Meanwhile, T2D is diagnosed based on any of the following criteria: fasting glucose levels of ≥5.6 mmol/L (100 mg/dL), 2 h glucose levels of ≥7.8 mmol/L (140 mg/dL) during an oral glucose tolerance test, HbA1c levels of ≥5.7% (30 mmol/mol), or a prior diagnosis of T2D. Similarly, hypertension is defined as blood pressure readings of ≥130/85 mmHg or the use of antihypertensive medications. Hypertriglyceridemia is characterized by plasma triglycerides of ≥1.7 mmol/L (150 mg/dL) or the use of lipid-lowering treatments. Finally, low HDL cholesterol is defined as HDL levels ≤1 mmol/L (40 mg/dL) in men or ≤1.4 mmol/L (50 mg/dL) in women, or the use of lipid-lowering medications [1,2].

The transition from NAFLD to MASLD represents a significant shift in hepatology, refining diagnosis, clinical practice, and treatment strategies. These changes necessitate increased early screening, particularly in high-risk populations like T2D patients, and reinforce the role of primary care physicians, endocrinologists, and cardiologists in disease detection. The diagnostic criteria now require hepatic steatosis with at least one cardiometabolic risk factor, shifting the focus from alcohol exclusion to metabolic dysfunction. Non-invasive tests such as FIB-4 index and elastography are emphasized over traditional liver biopsies, and comprehensive workups include viral hepatitis panels, autoimmune markers, and iron studies to exclude other liver diseases. This reclassification also reshapes treatment paradigms, prioritizing metabolic interventions alongside hepatic management [6,7].

Importantly, studies indicate that the prevalence of MASLD closely mirrors that of NAFLD, demonstrating that the updated terminology effectively captures most affected patients [2,8]. By 2030, it is projected that 33.5% of adults globally will have MASLD, with systematic reviews estimating the prevalence could reach 55.7% by 2040 [1,9].

Among individuals with T2D, the prevalence of MASLD is particularly striking, estimated at 65.3%. Over the years, this figure has increased significantly, rising from 55.8% between 1990 and 2004 to 68.8% during 2016–2021 [9]. While obesity and T2D are the primary risk factors, genetic and environmental influences also play a role. For example, MASLD prevalence stands at 57% in T2D patients with a normal BMI compared to 73% in those with obesity [9,10]. In the United States, MASLD prevalence is disproportionately high among Hispanic populations, older adults, and men. Data from the 2017–2018 National Health and Nutrition Examination Survey reveal that the prevalence of MASLD was 42.4%, with Hispanics showing the highest rates. However, no significant differences in advanced fibrosis rates were observed across racial or ethnic groups [4,11].

The progression of MASLD to metabolic-dysfunction-associated steatohepatitis (MASH)—previously termed non-alcoholic steatohepatitis (NASH)—marks a significant milestone in disease severity. MASH is characterized not only by lipid accumulation in the liver but also by the presence of inflammation and cellular damage [1,11,12].

Although MASH is often considered asymptomatic, nonspecific symptoms such as fatigue, right upper quadrant pain, and abnormalities in liver enzyme levels may begin to emerge. As MASH progresses, fibrosis develops, marked by the accumulation of scar tissue in the liver. If fibrosis becomes extensive, it can lead to cirrhosis, characterized by severe liver damage [12,13]. Advanced stages of cirrhosis may result in complications such as liver failure, hepatocellular carcinoma, and other life-threatening conditions. Notably, the progression of the disease varies significantly among patients. While some individuals may rapidly advance to cirrhosis, others may remain in earlier stages for many years. With appropriate treatment and management, the disease can even be reversed in some cases [12].

MASLD is associated with a wide range of cardiovascular and oncological effects. In terms of cardiovascular outcomes, MASLD has been linked to an increased risk of developing certain cardiovascular diseases. A meta-analysis revealed that individuals with MASLD have a 37% higher risk of cardiovascular events compared to those without the condition. Additionally, an association was observed between increased carotid intima-media thickness and the presence of high-risk atherogenic plaques [14,15].

MASLD and MASH are increasingly recognized as central components of the broader cardiometabolic disease spectrum. The recently proposed Cardiovascular–Renal–Hepatic–Metabolic syndrome expands upon the earlier Cardiovascular–Kidney–Metabolic model by integrating the liver as a critical organ involved in the interconnected pathophysiology linking metabolic, cardiovascular, renal, and hepatic health. This holistic framework acknowledges the bidirectional interactions between MASLD and other cardiometabolic disorders, emphasizing shared mechanisms such as meta-inflammation, adipokine dysregulation, insulin resistance, oxidative stress, and neurohormonal activation. By incorporating MASLD into the broader cardiometabolic landscape, the CRHM syndrome offers a clinically relevant and scientifically grounded framework to address the global burden of cardiometabolic disorders more effectively [16].

Regarding oncological outcomes, MASLD has also been associated with an elevated risk of both hepatic and extrahepatic cancers. While the primary hepatic cancers are often driven by cirrhosis, an increased risk has also been observed in patients with the condition. Furthermore, MASLD has also been linked to extrahepatic malignancies, particularly colorectal cancer, where the risk is approximately 50% higher compared to patients without MASLD [14,17].

Understanding the molecular pathophysiology and diverse clinical manifestations of these conditions is essential for effective treatment. Early diagnosis and targeted molecular interventions addressing the underlying mechanisms can halt disease progression and significantly improve patient outcomes.

## 2. Molecular Physiology

The accumulation of lipids in hepatocytes, a process known as hepatic steatosis, is a multifactorial phenomenon involving an imbalance among fatty acid uptake, synthesis, oxidation, and export. The following sections describe the key molecular mechanisms contributing to this process and their relevance in MASLD (Table 1).

### 2.1. Fatty Acid Uptake: Key Transporters

Fatty acid transporters, such as Cluster Differentiation 36 (CD36) and Fatty Acid Transport Protein 2 (FATP2), regulate hepatic fatty acid uptake. CD36 facilitates triglyceride accumulation by promoting Sterol Regulatory Element-Binding Protein 1 (SREBP1) activation through its interaction with *Insulin-Induced Gene 2* (INSIG2) [18]. Conversely, FATP2, an insulin-stimulated protein, converts fatty acids into acyl-CoA, which is subsequently utilized in triglyceride synthesis [19]. Indeed, Kupffer cells are responsible for increasing the expression of CD36 in response to a high-fat diet, highlighting their crucial role in the regulation of lipid metabolism [20].

Another key protein involved in fatty acid uptake is the Peroxisome Proliferator-Activated Receptor Alpha (PPARα). As a transcription factor, PPARα regulates genes associated with lipid uptake and metabolism, such as Carnitine Palmitoyltransferase 1 (CPT1), Acyl-CoA Oxidase 1 (ACOX1), CD36, and Fatty Acid Binding Protein 1 (FABP1). These genes are essential for the uptake, intracellular trafficking, and proper distribution of lipids within the cell, ensuring their channeling into appropriate metabolic pathways [21].

Another study emphasizes Sirtuin 6 (SIRT6) as a negative regulator of fatty acid uptake. Its deficiency promotes the expression of fatty acid transporters such as CD36, FATP2, and FABP1, facilitating lipid entry into hepatocytes and contributing to hepatic fat accumulation. This suggests that SIRT6 normally acts as a brake on excessive lipid uptake in the liver, promoting a more balanced metabolism. This effect is thought to occur through the repression of PPARγ, a transcription factor that regulates the expression of fatty acid transporter genes [22].

Fibroblast Growth Factor 21 (FGF21) appears to modulate fatty acid uptake by influencing the expression of CD36, a key transporter that facilitates lipid entry into hepatocytes. This regulatory mechanism suggests that, under normal conditions, FGF21 prevents excessive fatty acid absorption, thereby protecting the liver from unnecessary fat accumulation [23] (Figure 1).

METTL3 plays a critical role in hepatic lipid homeostasis by acting as a negative regulator of excessive lipid uptake. METTL3 represses the expression of CD36, thereby limiting fatty acid entry into hepatocytes and preventing unnecessary lipid accumulation that could lead to lipotoxicity. This regulation occurs through epigenetic modifications, as METTL3 associates with histone deacetylases HDAC1/2 to induce the deacetylation of H3K9 and H3K27 at the Cd36 promoter, reducing its transcription. In terms of lipid transport, METTL3 likely contributes to a balance between lipid uptake, oxidation, and secretion in the form of lipoproteins [24].

### 2.2. Lipogenesis: Synthesis of New Lipids

Lipogenesis is a pivotal process in triglyceride accumulation, primarily regulated by the transcription factor SREBP1, which governs the expression of lipid synthesis-related genes, including Fatty Acid Synthase (FAS), Stearoyl-CoA Desaturase-1 (SCD-1), and Acetyl-CoA Carboxylase (ACC1). Upon activation by insulin and glucose, SREBP1 significantly enhances the number and size of lipid droplets in hepatocytes, thereby promoting lipid accumulation [25,26]. Furthermore, the enzyme N-Acetyltransferase 10 (NAT10) has been shown to stabilize SREBP1c mRNA through acetylation, which further amplifies the de novo lipogenesis and contributes to hepatic lipid accumulation [27].

In addition to SREBP1, other molecular markers, such as FAS, play a critical role in lipid synthesis and insulin resistance. FAS has emerged as a potential biomarker for insulin resistance and type 2 diabetes [28]. Insulin resistance disrupts not only lipid synthesis but also key insulin signaling pathways, such as protein kinase B (AKT) phosphorylation. Under normal physiological conditions, phosphorylated AKT inhibits Glycogen Synthase Kinase 3 Beta (GSK-3β), thereby promoting glycogen synthesis and improving insulin sensitivity. However, in states of insulin resistance, AKT phosphorylation is impaired, leading to the persistent activation of GSK-3β, which contributes to elevated blood glucose levels [29].

Interestingly, the inhibition of GSK-3β has been shown to attenuate lipotoxicity, underscoring its role in metabolic dysfunction [30]. When AKT fails to phosphorylate GSK-3β, the latter remains active, exacerbating lipotoxicity and further disrupting lipid metabolism and glucose regulation. These findings highlight the interconnected nature of insulin signaling, lipid metabolism, and glucose homeostasis, suggesting that therapeutic strategies targeting these pathways could mitigate metabolic dysfunction.

In vitro studies using primary human liver cultures and HepG2 cells have provided additional insights into these mechanisms. These studies demonstrated that insulin-induced de novo lipogenesis activates SREBP1, ATP Citrate Lyase (ACLY), and acetyl-CoA carboxylase alpha (ACCα), leading to the accumulation of lipid droplets in hepatocytes. This activation underscores the critical role of SREBP1 in driving the expression of enzymes necessary for lipid synthesis and its subsequent impact on metabolic health [18].

### 2.3. Fatty Acid Oxidation: A Metabolic Counterbalance

Fatty acid oxidation serves as a vital mechanism for preventing lipid accumulation and maintaining energy homeostasis. This process is primarily regulated by Peroxisome Proliferator-Activated Receptor Alpha (PPARα), a nuclear transcription factor that activates genes involved in lipid oxidation across mitochondria, peroxisomes, and the endoplasmic reticulum [31]. By promoting the breakdown of fatty acids, PPARα plays a crucial role in reducing lipid accumulation and counterbalancing lipogenesis.

In addition to its role in lipid metabolism, PPARα exerts significant anti-inflammatory effects. It achieves this by indirectly inhibiting nuclear factor-kappa B (NF-κB), a key regulator of inflammatory pathways. This dual function underscores the importance of PPARα activity in maintaining both the metabolic and inflammatory balance [32]. However, when the PPARα expression is epigenetically downregulated, such as through promoter hypermethylation, its protective effects are diminished. This downregulation exacerbates inflammation, impairs lipid oxidation, and contributes to the development of metabolic dysfunction [33].

Another key player in fatty acid oxidation is the protein Long-Chain Acyl-CoA Synthetase Family Member 5 (ACSL5), which facilitates the activation and subsequent oxidation of long-chain fatty acids. ACSL5 activity is regulated by Ubiquitin-Specific Protease 29 (USP29), which prevents its degradation and ensures its functional stability. However, a decrease in USP29 levels, as observed in hepatocytes exposed to high-fat diets, disrupts this regulatory mechanism. Reduced USP29 leads to impaired ACSL5 activity, which, in turn, worsens insulin resistance, promotes hepatic steatosis, and intensifies inflammation [34].

These interconnected pathways highlight the importance of PPARα and ACSL5 in fatty acid oxidation and underscore their roles in maintaining metabolic equilibrium. Their dysfunction not only impairs lipid metabolism but also amplifies inflammatory and insulin resistance pathways, further aggravating metabolic disorders.

### 2.4. Safe Storage of Lipid Droplets: Droplets of Hepatic Lipid Balance

The safe storage of triglycerides in lipid droplets is critical for preventing lipotoxicity and maintaining cellular homeostasis. Lipid droplets are stabilized by structural proteins such as Perilipin 2 (PLIN2) and Diacylglycerol Acyltransferase 2 (DGAT2), which are essential for their formation and function. DGAT2 catalyzes the final step in triglyceride synthesis, a process tightly regulated by insulin to ensure proper energy storage [35,36,37].

Recent evidence suggests that the transcription factor p63, a member of the p53 family, plays a pivotal role in lipid accumulation within hepatocytes. Specifically, the TAp63 isoform has been shown to induce lipid droplet formation in mouse models by inhibiting fatty acid oxidation. This inhibition is mediated through the upregulation of Autophagy-related 3 (ATG3), a protein whose expression increases in response to elevated TAp63 levels. Notably, TAp63 expression is significantly upregulated under high-fat diet conditions, further driving lipid storage and metabolic dysregulation [38].

In addition, Glutathione S-Transferase A1 (GSTA1) has emerged as a critical player in lipid droplet formation within hepatocytes. Studies conducted in cell cultures and human tissue samples diagnosed with MASLD have revealed that low levels of GSTA1 are associated with lipid and triglyceride accumulation. Under normal physiological conditions, GSTA1 promotes the degradation of Fatty Acid Binding Protein 1 (FABP1), thereby limiting fatty acid uptake and synthesis. However, reduced GSTA1 levels disrupt this regulatory mechanism, leading to enhanced lipid accumulation and contributing to metabolic dysregulation [39].

Moreover, Peroxisome Proliferator-Activated Receptor Gamma Coactivator 1-Alpha (PGC1α) has been identified as a key regulator of mitochondrial function and lipid metabolism. Intriguingly, PGC1α expression increases in the intestines of mice fed a Western diet, promoting lipid synthesis through the regulation of hepatic lipid metabolism. The targeted deletion of PGC1α in the intestine significantly reduces the expression of lipogenic genes, including Srebp1c, Fasn, and Scd1, and protects against hepatic lipid accumulation. This protective effect is mediated through the modulation of the Liver X Receptor (LXR), a central regulator of lipid and cholesterol metabolism. These findings highlight the influence of intestinal PGC1α on hepatic lipid accumulation and its potential as a therapeutic target [40].

Lipid accumulation in the liver is also closely linked to inflammation, with the chemokine C-X3-C Motif Ligand 1 (CX3CL1) and its receptor C-X3-C Motif Receptor 1 (CX3CR1) playing key roles in macrophage migration and polarization. In mouse models, M1 macrophages and hepatic stellate cells have been identified as primary expressors of CX3CL1 and CX3CR1, contributing to inflammatory responses. Importantly, the absence of CX3CR1 shifts the balance toward the pro-inflammatory C-C Motif Chemokine Ligand 2 (CCL2)/C-C Motif Chemokine Receptor 2 (CCR2) axis, which exacerbates macrophage infiltration and inflammation. This underscores the critical role of CX3CL1/CX3CR1-mediated signaling as a counter-regulatory mechanism that limits excessive inflammation and slows the progression of MASH [41].

In conclusion, lipid droplet formation and storage are tightly regulated by a network of proteins and signaling pathways that ensure metabolic balance. The dysregulation of these mechanisms not only contributes to lipid accumulation but also amplifies inflammatory processes, underscoring their relevance in the pathogenesis of metabolic disorders.

### 2.5. Lipid Exportation: Pathways of Hepatic Lipid Clearance

The Microsomal Triglyceride Transfer Protein (MTTP) plays a crucial role in lipid metabolism by catalyzing the assembly of triglyceride-rich lipoproteins, such as very low-density lipoproteins (VLDL), in the liver. Primarily located in the endoplasmic reticulum, MTTP functions as a lipid chaperone, facilitating the transfer of triglycerides, cholesterol, and phospholipids to apolipoproteins, such as Apolipoprotein B-100 (ApoB-100) [42]. This process is essential for the formation of functional lipoprotein particles required to export lipids from the liver as VLDL [43].

SIRT6 deacetylates ACSL5, a key enzyme in the activation of long-chain fatty acids, thereby facilitating its oxidation and preventing lipid accumulation in the liver. At the molecular level, the transport and metabolism of lipids in the liver are regulated by various proteins and enzymes that control the balance between fatty acid availability, their storage as triglycerides, and their elimination through oxidation or the secretion of VLDL. In this context, ACSL5 plays a crucial role in the conversion of free fatty acids into acyl-CoA, facilitating their subsequent metabolism. The deacetylation of ACSL5 by SIRT6 enhances its activity in fatty acid oxidation, thereby reducing hepatic lipid accumulation [44].

Transmembrane protein 41B (TMEM41B) facilitates the movement of phospholipids across the endoplasmic reticulum (ER) membrane bilayer, enabling the formation of lipoproteins that transport lipids through the bloodstream. TMEM41B deficiency in the liver leads to an almost complete elimination of plasma lipids due to the absence of mature lipoproteins in the ER, which paradoxically also activates lipid synthesis in the liver. VLDL and chylomicrons are assembled in the ER and Golgi before being secreted into the circulation. In this process, TMEM41B plays a fundamental role in the movement of phospholipids from the cytoplasmic monolayer of the ER bilayer to the luminal monolayer, ensuring the proper coating and stability of forming lipoproteins. Additionally, TMEM41B interacts with Surfeit Locus Protein 4 and Apolipoprotein B (ApoB), a key structural component of lipoproteins, suggesting that TMEM41B is an essential regulator of lipoprotein assembly [45].

Apolipoprotein B is a key protein in the formation of very low-density lipoproteins in the liver and chylomicrons in the intestine. In its full-length form, ApoB-100 is essential for the secretion of VLDL and its conversion into LDL, while the truncated version, ApoB-48, participates in the formation of chylomicrons in the intestine. During VLDL synthesis in the liver, ApoB-100 associates with triglycerides and other lipids through its interaction with microsomal triglyceride transfer protein. This interaction is crucial as it allows ApoB to fold correctly and stabilize with an adequate lipid load, preventing its premature degradation in the ER. Once assembled, VLDL is secreted into the bloodstream, where it is metabolized to form LDL, which transports cholesterol to cells. If ApoB-100 is not synthesized correctly or cannot interact with microsomal triglyceride transfer protein, as occurs in certain mutations, VLDL secretion is drastically reduced, leading to lipid accumulation in the liver [46].

Activating transcription factor 3 (ATF3), when overexpressed in hepatocytes, promotes hepatic uptake of HDL, inhibits intestinal absorption of fat and cholesterol, and enhances reverse cholesterol transport in macrophages. These effects are achieved through the induction of scavenger receptor class B type 1 and the repression of cholesterol 12α-hydroxylase, impacting bile acid synthesis and cholesterol excretion. ATF3 in the liver is inhibited by hydrocortisone, a stress hormone, suggesting a link between chronic stress and the regulation of lipid metabolism and atherosclerosis. The reduced ATF3 expression in the liver is associated with elevated plasma HDL levels, indicating that its downregulation may be related to cholesterol accumulation in the circulation. ATF3 regulates hepatic HDL uptake through its interaction with p53, a tumor suppressor that also regulates the induction of scavenger receptor class B type 1 expression. Furthermore, ATF3 modulates lipid absorption in the intestine through its interaction with hepatic nuclear factor 4α, resulting in the inhibition of cholesterol 12α-hydroxylase expression, thereby reducing cholic acid production and affecting bile acid reabsorption in the intestine [47].

### 2.6. Role of Adipokines in the Pathogenesis and Progression of MASLD/MASH

An increasing body of evidence demonstrates that individuals with MASLD and its progressive form, MASH, exhibit significant alterations in adipokine profiles compared to healthy individuals. Adipokines, which are bioactive molecules secreted primarily by adipose tissue, play a crucial role in the pathogenesis of MASLD by influencing metabolic processes, insulin sensitivity, and inflammatory pathways within the liver. In obesity, the production of pro-inflammatory adipokines is markedly elevated, contributing to a chronic state of low-grade inflammation that accelerates hepatic injury and fibrosis. However, not all adipokines exert detrimental effects; certain adipokines possess anti-inflammatory properties and may counteract liver inflammation and metabolic dysfunction. The imbalance between pro- and anti-inflammatory adipokines is now recognized as a key mechanism driving the transition from hepatic steatosis to more severe forms of liver disease, including MASH [48].

Among these, adiponectin has been shown to enhance fatty acid oxidation, increase insulin sensitivity, and suppress pro-inflammatory cytokines through the activation of the AMPK/JNK/ERK1/2 pathway, while also reducing oxidative stress by lowering the malondialdehyde and NOX2 levels and improving mitochondrial function via the upregulation of UCP2 [49,50]. In contrast, leptin, despite its beneficial role in lipid mobilization, contributes to hepatic oxidative stress and Kupffer cell activation under conditions of metabolic dysfunction, fostering inflammation and fibrosis [51]. FGF21, an endocrine hormone secreted by both the liver and adipose tissue, exerts hepatoprotective effects by activating the AMPK-SIRT1 and PI3K/Akt pathways, enhancing antioxidant capacity, mitochondrial β-oxidation, and insulin sensitivity, while also mitigating ER stress [52,53]. Other adipokines, such as RBP4, promote hepatic steatosis by increasing pro-inflammatory cytokine release, de novo lipogenesis, and mitochondrial lipid oxidation dysfunction [54,55], whereas irisin has shown protective effects by alleviating oxidative stress, reducing lipid accumulation, and promoting mitochondrial biogenesis [56,57]. Furthermore, chemerin, vaspin, and apelin exhibit anti-inflammatory, insulin-sensitizing, and anti-apoptotic effects [58,59,60], whereas resistin and visfatin contribute to insulin resistance, ER stress, and inflammatory cytokine production, exacerbating hepatic injury [61,62]. Additional protective roles have been identified for IGF-1, adipsin, and omentin-1, which collectively reduce oxidative stress, enhance mitochondrial function, and suppress apoptosis [63,64,65]. Conversely, IL-6 has been implicated in promoting mitochondrial dysfunction and apoptosis, further accelerating MASLD progression [66]. This diverse functional spectrum highlights the intricate and often opposing roles of adipokines in shaping the pathophysiological landscape of MASLD and MASH.

### 2.7. Inflammation and Oxidative Stress: Drivers of Liver Injury and Fibrosis

Inflammation and oxidative stress are pivotal drivers in the progression from MASLD to MASH, intricately linking metabolic dysregulation with immune responses and fibrogenesis. Pro-inflammatory cytokines, such as CX3CL1 and interleukin (IL) 11, significantly contribute to hepatocyte injury and fibrosis by activating both autocrine and paracrine signaling pathways. These pathways impair mitochondrial function, promote lipid accumulation, and drive the activation of hepatic stellate cells, perpetuating the fibrotic process [67]. Similarly, the downregulation of key regulatory proteins exacerbates inflammation. For instance, the reduced expression of Suppressor of Cytokine Signaling 2 (SOCS2) facilitates the activation of NF-κB and inflammasome pathways, while diminished levels of Metallothionein-1B (MT1B) enhance pro-inflammatory cytokine production and fibrosis via the Phosphoinositide 3-kinase (PI3K)/AKT signaling pathway [68,69].

Additionally, Kupffer cell activation plays a central role in perpetuating inflammation. This process is mediated by mitochondrial DNA released from damaged hepatocytes, which engages the Toll-like receptor (TLR) 9 and Stimulator of Interferon Genes (STING) signaling pathways [70]. Lipid overload and oxidative stress further exacerbate these inflammatory cascades by generating elevated levels of reactive oxygen species (ROS). These ROS impair mitochondrial β-oxidation and deplete critical antioxidants, such as mitochondrial glutathione, creating a feedback loop that exacerbates hepatic damage [71,72]. In experimental models of MASH, reduced glutathione levels and increased lipid peroxidation have been observed. Notably, caffeine administration has demonstrated protective effects by modulating the TLR4/Mitogen-activated protein kinase (MAPK) pathway, inhibiting Jun N-terminal kinase (JNK), extracellular signal-regulated kinase (ERK), and p38 activation. This intervention reduces the production of pro-inflammatory cytokines, including IL-17, IL-1β, and tumor necrosis factor-alpha (TNF-α), while also preventing inflammasome activation [73].

Proteins with anti-inflammatory and antioxidant properties, such as Fibronectin Type III Domain Containing Protein 4 (FNDC4) and glutathione peroxidase 7 (GPx7), play protective roles against MASLD progression. However, their reduced expression exacerbates inflammation and oxidative stress, accelerating disease progression [74,75]. Furthermore, lipotoxicity induced by a high-fat diet (HFD) contributes to hepatic stellate cell loss, triggering macrophage recruitment to compensate for the loss of Kupffer cells in the toxic microenvironment. The oxidative stress-related Chemokine-like receptor 1 (CMKLR1), is also downregulated in HFD-fed mice, resulting in a MASH phenotype characterized by elevated TNF-α and IL-6 levels, reduced superoxide dismutase (SOD) activity, and impaired autophagy. Intriguingly, chemerin administration mitigates these effects by activating the Janus kinase (JAK)2/signal transducer and activator of transcription (STAT) 3 pathway, reducing inflammation and oxidative damage [76,77,78,79].

Moreover, the intestinal deletion of PGC1α has been shown to decrease inflammatory markers such as CCL2 and TNF-α, as well as the M1/M2 macrophage ratio, while also reducing the TGF-β expression, a key activator of hepatic stellate cells [40]. HFD models further emphasize the role of transcriptional regulators such as Forkhead box protein O1 (FOXO1), Yes-associated protein (YAP), and Neurogenic locus notch homolog protein 1 (NOTCH1). These regulators drive inflammatory cascades through the cGAS-STING signaling pathway, promoting macrophage polarization, and enhancing cytokine release [80,81]. Collectively, these interconnected mechanisms highlight the intricate interplay between inflammatory and oxidative stress pathways in accelerating hepatic fibrosis and dysfunction, underscoring their relevance as potential therapeutic targets to mitigate the progression of MASLD to MASH.

### 2.8. Mitochondrial Dysfunction: The Energy Crisis in MASLD

Mitochondrial dysfunction plays a pivotal role in the progression of MASLD and its advanced stages, such as MASH, as it is driven by metabolic stress, lipid overload, and inflammation. Under these conditions, hepatocyte mitochondria become overwhelmed by the excessive accumulation of lipids, leading to impaired energy production, a reduced oxidative capacity, and increased ROS generation. Consequently, this initiates a vicious cycle characterized by oxidative stress, lipid peroxidation, the release of inflammatory cytokines, and hepatocyte apoptosis. Together, these processes exacerbate hepatic inflammation and fibrosis, further driving disease progression [82]. Moreover, compromised mitochondrial function manifests as diminished ATP production, reduced basal and maximal respiratory capacity, and decreased energy reserves. Additionally, the activity of β-hydroxyacyl-CoA dehydrogenase (β-HAD), a critical enzyme involved in β-oxidation, is significantly reduced, further impairing lipid metabolism and energy homeostasis [83]. These interconnected mechanisms highlight the central role of mitochondrial dysfunction in the pathogenesis of MASLD and its more severe forms.

Mitochondrial dysfunction is intricately linked to impaired autophagy and mitophagy, which are critical processes for maintaining cellular homeostasis. In lipid-overloaded hepatocytes, proteins such as DDX58 and TMEM55B, which regulate autophagy and lysosomal trafficking, are significantly downregulated. This downregulation leads to autophagic dysfunction, resulting in the accumulation of damaged mitochondria and pronounced mitochondrial fission [84,85]. Furthermore, abnormalities in mitochondrial dynamics, including a decreased expression in Dynamin-related protein 1 (Drp1) and Optic Atrophy 1 (OPA1), impair the essential processes of mitochondrial fission and fusion. These disruptions contribute to the formation of megamitochondria that are resistant to mitophagy, further exacerbating mitochondrial dysfunction [86,87].

In addition to these structural impairments, reduced mitochondrial biogenesis plays a pivotal role in limiting mitochondrial renewal and metabolic resilience. This is evidenced by the decreased expression of key regulators such as PGC1α, AMP-activated protein kinase (AMPK), and silent information regulator 1 (SIRT1), which are essential for mitochondrial maintenance and energy homeostasis [83]. The inflammatory microenvironment, particularly the elevated levels of cytokines like TNF-α, further aggravates mitochondrial damage by impairing mitochondrial DNA integrity, disrupting respiratory function, and reducing the activity of the oxidative phosphorylation complexes [74].

In HFD mouse models, mitochondria exhibited abnormal morphologies, including the vacuolization and disorganization of mitochondrial cristae. These alterations are associated with a significant reduction in ATP levels, highlighting a failure in the cellular energy capacity. Moreover, the inhibition of mitophagy, a key process for the removal of damaged mitochondria, exacerbates this dysfunction. Proteins such as BCL2/adenovirus E1B 19 kDa-interacting protein 3 (BNIP3) and p62, which are essential for the regulation of mitophagy, show a decreased expression in hepatocytes affected by MASLD/MASH, leading to the accumulation of defective mitochondria and an increase in cellular apoptosis [88].

Additionally, the protein Acylglycerol Kinase (AGK) is found to be reduced in the livers of patients with MASH, and its knockout in mice results in MASH development. AGK is a subunit of the Translocase of the Inner Mitochondrial Membrane 22 (TIM22) complex, responsible for the import of mitochondrial proteins into the inner membrane. AGK deficiency leads to the loss of mitochondrial cristae, a decrease in ATP production, and impaired mitochondrial respiration [89].

Emerging therapeutic interventions offer some promise in mitigating these issues. For instance, the addition of FNDC4 or targeting proteins such as AGK, which is critical for mitochondrial protein import and cristae integrity, has shown potential in alleviating mitochondrial dysfunction and partially restoring metabolic function. However, despite these advances, achieving complete recovery remains a significant challenge [74,89].

Collectively, these findings underscore the central role of mitochondrial dysfunction in the progression of MASLD and highlight the importance of developing therapeutic strategies that focus on restoring mitochondrial homeostasis.

### 2.9. Endoplasmic Reticulum Stress: A Cellular Stressor in Liver Disease

Endoplasmic reticulum stress plays a pivotal role in exacerbating hepatic lipid accumulation and inflammation in MASLD. Specifically, endoplasmic reticulum stress upregulates the expression of very-low-density lipoprotein receptors (VLDLRs) through a pathway mediated by Activating Transcription Factor 4 (ATF4), thereby promoting increased lipid uptake and storage in hepatocytes. This process is further aggravated by high fructose consumption, which not only elevates VLDLR levels but also reduces the expression of SIRT1, a protective regulator that mitigates steatosis. SIRT1 exerts its protective effects by attenuating NF-κB signaling, thereby suppressing hepatic inflammation and improving metabolic homeostasis [90].

In addition to endoplasmic reticulum stress, gut-microbiota-derived metabolites also contribute to hepatic lipid dysregulation. For instance, metabolites like 3-hydroxyphenylacetic acid, produced by *Phocaeicola vulgatus*, indirectly modulate hepatic lipid accumulation by inhibiting key enzymes involved in steroid synthesis. This highlights the complex interplay between gut microbiota and liver metabolism in MASLD [91].

Lipid overload further triggers significant inflammatory responses. Studies on primary hepatocyte cultures subjected to lipotoxic stress have demonstrated that the increased expression of Bromodomain-containing protein 4 (BRD4) enhances the acetylation of histone H3 at lysine 27 in the promoter region of Gasdermin D. This epigenetic modification, in conjunction with inflammasome activation, drives pyroptosis and the subsequent release of pro-inflammatory cytokines, including IL-1β, monocyte chemoattractant protein-1 (MCP-1), and TNF-α. The inflammatory cascade is amplified by the NLRP3 inflammasome, which interacts with heat shock protein 70 (Hsp70). This interaction facilitates the release of Hsp70 as a damage-associated molecular pattern (DAMP), which binds to TLR4 receptors on neighboring hepatocytes. The binding activates the NF-κB pathway, perpetuating a pro-inflammatory cascade that exacerbates liver injury [92].

Moreover, lipotoxicity intensifies endoplasmic reticulum stress, which synergistically promotes both pyroptosis and NLRP3 inflammasome activation. This creates a vicious cycle of inflammation and hepatocyte damage, further contributing to the progression of MASLD. Collectively, these interconnected mechanisms highlight the critical role of endoplasmic reticulum stress, lipid overload, and gut microbiota in driving the pathogenesis of MASLD [93].

### 2.10. Gut Microbiota Dysbiosis: The Gut–Liver Axis in MASLD Progression

The interplay between gut microbiota dysbiosis, lipopolysaccharides (LPS), and inflammatory pathways plays a pivotal role in the progression of MASLD. An imbalance among bacterial phyla, including Verrucomicrobiae, Bacteroidetes, Proteobacteria, and Cyanobacteria, contributes to increased levels of LPS and pro-inflammatory cytokines such as TNF-α and IL-6. These inflammatory mediators activate critical signaling pathways, including TLR4/NF-κB and AMPK, thereby promoting hepatic steatosis and inflammation. Notably, dysbiosis induced by a cocoa-paste-based diet has been shown to increase levels of Bacteroidetes, Proteobacteria, and Firmicutes, which correlate with MASLD, insulin resistance, and obesity [94].

LPS, a potent endotoxin derived from gut bacteria, further exacerbates inflammation by driving IL-6 production via the TLR2/MyD88/IKKα/NF-κB pathway. This cascade amplifies the inflammatory response, contributing to the progression of liver damage and metabolic dysfunction [95,96].

Kupffer cells, the resident macrophages of the liver, are particularly sensitive to changes in the gut microbiota. Upon exposure to increased levels of LPS, they polarize toward the M1 pro-inflammatory phenotype, resulting in the elevated secretion of cytokines such as TNF-α, IL-1β, and IL-12, along with ROS. This polarization significantly amplifies hepatic inflammation and oxidative stress [97]. Moreover, intestinal dysbiosis exacerbates this process by increasing the intestinal permeability. This is mediated, in part, by the upregulation of serine-threonine kinase 39 (STK39), a negative regulator of intestinal barrier integrity, which facilitates the translocation of LPS into the systemic circulation [98].

LPS further amplifies inflammatory signaling through its interactions with downstream cytokine pathways. Specifically, TNF-α induces NF-κB activation, while IL-6 triggers the JAK-STAT3 signaling pathway. IL-6 binds to its receptor and activates JAK, which subsequently phosphorylates STAT3. The phosphorylated STAT3 translocates to the nucleus, enhancing the transcription of genes that sustain inflammation and metabolic dysfunction. This perpetual cycle of inflammatory signaling underscores the critical role of the gut–liver axis in driving hepatic inflammation, steatosis, and the progression of MASLD [99].

Collectively, these findings highlight the intricate connections between gut microbiota dysbiosis, LPS-mediated signaling, and pro-inflammatory pathways, emphasizing their central role in the pathogenesis of MASLD.

### 2.11. Unraveling the Spleen–Liver Axis

The liver and spleen are physiologically interconnected, with spleen volume changes reflecting liver pathology due to portal hypertension and fibrosis progression.

Recent studies highlight the significance of spleen volume as a biomarker in liver disease. Helgesson et al. analyzed over 37,000 individuals using MRI and found that spleen volume correlates positively with liver fat fraction, fibrosis score, and liver volume. This study suggests that spleen enlargement occurs early in MASLD progression, potentially due to increased portal pressure [100]. Furthermore, volumetric assessments of the liver and spleen via CT and MRI have been proposed as reliable, non-invasive biomarkers for diagnosing and staging liver fibrosis. These measurements provide valuable prognostic information, surpassing conventional diagnostic approaches such as ultrasound [101].

The spleen–liver axis has been further elucidated by Zhang et al. (2023), who demonstrated that splenic CD11b + CD43hiLy6Clo monocytes exacerbate liver fibrosis. These cells migrate from the spleen into the fibrotic liver, where they differentiate into macrophages that activate hepatic stellate cells and promote fibrogenesis. Single-cell RNA sequencing identified these spleen-derived monocytes as key mediators of inflammation and fibrosis, revealing potential therapeutic targets for controlling liver disease progression [102]. These findings underscore the clinical potential of spleen-derived biomarkers and immune cell modulation as novel strategies for early detection, monitoring, and therapeutic intervention in liver fibrosis and cirrhosis.

### 2.12. Genetic and Epigenetic Factors: Molecular Determinants of Disease Susceptibility

Genetic and epigenetic factors play a significant role in the pathogenesis and progression of MASLD, highlighting their impact on lipid metabolism, mitochondrial function, and fibrogenesis. Among genetic determinants, the Patatin-like phospholipase domain-containing protein 3 (PNPLA3) I148M variant is pivotal in promoting triglyceride accumulation. This mutation disrupts the dynamics between lipid droplets and the Golgi apparatus, facilitating lipid accumulation and inducing proteomic changes associated with inflammation. These alterations underscore the critical role of PNPLA3-I148M in the development and progression of MASLD [103].

On the epigenetic front, microRNAs such as miR-33, embedded within the SREBP2 gene, have emerged as central regulators in MASLD progression. Elevated levels of miR-33 in both the liver and serum are strongly associated with impaired mitochondrial function, disrupted lipid metabolism, and increased fibrogenesis. Experimental studies have demonstrated that the deletion of miR-33 significantly mitigates de novo lipogenesis, enhances fatty acid oxidation, and improves mitochondrial dynamics by upregulating key metabolic regulators, including CPT1α, PGC1α, and AMPKα. These epigenetic interventions not only reduce lipid and cholesterol accumulation but also alleviate hepatocyte ballooning and decrease fibrosis markers such as fibronectin, COL1A1, and hydroxyproline. Furthermore, the deletion of miR-33 promotes antifibrotic responses in hepatic stellate cells, underscoring its potential as a therapeutic target [104].

In addition to microRNA-mediated regulation, other epigenetic modifications, such as hypermethylation in the NADH dehydrogenase 6 (ND6) region of mitochondrial DNA, exacerbate mitochondrial dysfunction in MASLD. Hypermethylation disrupts the mitochondrial cristae structure, induces mitochondrial swelling, and leads to the basal hyperactivity of mitochondrial respiration, which collectively contribute to lipid accumulation. Furthermore, the hypermethylation of mitochondrial DNA influences the expression of nuclear genes involved in lipid and bile acid metabolism, thereby intensifying metabolic and structural alterations in the mitochondria. These compounded effects highlight the intricate link between mitochondrial dysfunction and MASLD progression [105].

In summary, genetic factors such as the PNPLA3-I148M variant, alongside epigenetic mechanisms including miR-33 and mitochondrial DNA hypermethylation, drive the metabolic, inflammatory, and fibrotic changes characteristic of MASLD. These insights emphasize the importance of targeting both genetic and epigenetic pathways in developing effective therapeutic strategies [104,105].

## 3. Clinic Manifestations

### 3.1. Early Stages of MASLD

MASLD has often been considered asymptomatic, particularly in its early stages. However, this does not imply that individuals with the condition experience no clinical manifestations or that the disease has no impact on quality of life. Research indicates that many patients with MASLD report a variety of symptoms, even in the absence of advanced fibrosis, which challenges the perception of MASLD as a silent disease [106,107,108].

MASLD typically develops gradually over several years, often without causing apparent symptoms in its early stages. When lipids initially begin to accumulate in the liver, patients may be asymptomatic or present with very mild symptoms. This lack of overt clinical signs has earned MASLD the reputation of being a “silent disease”, as its slow progression often allows the pathology to advance undetected for extended periods [12,109]. The liver’s remarkable functional reserve capacity further complicates the detection of MASLD. This capacity allows the liver to maintain relatively normal function despite significant cellular damage, masking symptoms even in advanced stages such as fibrosis or cirrhosis. As a result, diagnosing MASLD often becomes a significant challenge, with the disease frequently remaining unnoticed until substantial progression has occurred [106,109,110].

When symptoms are present, they are usually vague and nonspecific, including fatigue, discomfort in the right upper quadrant of the abdomen, and general malaise. These symptoms can easily be misattributed to other conditions, further complicating the diagnostic process [12,109,110,111].

Biochemical markers can provide some insight into early MASLD, though they are often insufficient for definitive diagnosis. For example, alanine aminotransferase (ALT) and aspartate aminotransferase (AST) levels, commonly used to assess liver function, may remain within normal ranges or show only slight elevations in early MASLD. This is because significant inflammation or cellular damage, which would cause more pronounced changes in these enzymes, is not typically present at this stage [110,112,113,114]. Similarly, gamma-glutamyl transferase (GGT) levels may be altered in MASLD but are more likely to be elevated in advanced stages with significant inflammation. Simple steatosis, a hallmark of early MASLD, is less likely to result in elevated GGT levels [115]. Another biochemical marker, alkaline phosphatase (ALP), which is found in various tissues, including the liver, bones, and intestines, may also indicate liver or bile duct damage when elevated. However, like GGT, ALP levels are more commonly elevated in advanced MASLD stages characterized by inflammation and fibrosis [115].

While MASLD often progresses silently, early-stage symptoms such as fatigue and sleep disturbances, combined with subtle biochemical changes, may provide important clues for its identification. Recognizing these early manifestations is crucial for timely diagnosis and intervention, potentially preventing the progression to more severe stages of the disease.

### 3.2. The Silent Progression of MASLD (MASH)

MASH, particularly in its early stages, is often regarded as a “silent” disease due to its asymptomatic nature. However, numerous studies emphasize the critical importance of early detection and effective management to prevent severe complications and enhance patients’ quality of life [12,116]. Among the various pathological hallmarks of MASLD, fibrosis represents a pivotal turning point in the disease’s natural history. This is because fibrosis is strongly associated with an increased risk of complications and mortality [110,117].

The global burden of MASLD has escalated to epidemic proportions, with its prevalence rising continuously, especially in Western countries [118,119]. This increasing prevalence, coupled with the potential for progression to severe liver disease, places a substantial strain on healthcare systems worldwide. Consequently, early detection and the adoption of preventive strategies are essential for mitigating the growing challenges posed by this condition effectively.

Another frequently reported symptom in MASH is pain in the upper right quadrant of the abdomen, which affects approximately 61% of patients. This pain is typically described as a persistent, dull ache that may radiate to the back. Some patients also experience abdominal distension, further impacting their daily comfort and quality of life [109,120]. In addition, a generalized sense of malaise characterized by weakness, loss of appetite, and nausea may emerge. Cognitive impairments, such as difficulty concentrating and memory loss, are also common in MASH. A study using an animal model linked cognitive dysfunction and depression-like behavior to systemic inflammation and neuroinflammation [119]. These cognitive deficits can impair daily functioning, strain social relationships, reduce mood, and increase anxiety levels [109].

The early recognition of MASLD and T2D, coupled with a thorough understanding of their complex interplay, is essential for implementing timely and effective interventions to halt disease progression. Robust evidence demonstrates that T2D exacerbates the development and progression of MASLD, emphasizing the need for future research to prioritize the aggressive prevention and targeted treatment of T2D as a key strategy for improving MASLD outcomes [121]. Furthermore, by understanding the silent progression of MASH and its diverse clinical manifestations, healthcare providers can enhance early diagnosis and deploy tailored interventions. These measures are critical not only for improving patient outcomes but also for alleviating the growing burden of MASH on healthcare systems.

### 3.3. Screening and Diagnostic Approach for MASLD

Screening for MASLD is particularly important in patients with metabolic risk factors, including obesity, type 2 diabetes, hypertension, dyslipidemia, and metabolic syndrome. Among these, individuals with type 2 diabetes are at especially high risk, with a prevalence reaching up to 65% in this population. Initial screening should include liver function tests (LFTs) and the calculation of the FIB-4 index, a widely validated and simple tool for assessing fibrosis risk. However, FIB-4 has limitations in younger patients under 35 years and older adults over 65 years, where it tends to underestimate or overestimate fibrosis risk, respectively. In some cases, alternative scores such as the NAFLD Fibrosis Score (NFS) or the Enhanced Liver Fibrosis (ELF) test may provide additional value, particularly for intermediate-risk patients or when a higher accuracy in detecting advanced fibrosis is required. Imaging also plays a crucial role in MASLD evaluation. Abdominal ultrasound, while useful for detecting hepatic steatosis, offers limited utility for fibrosis assessment. Therefore, elastography (such as FibroScan) is preferred in patients with elevated FIB-4 or suspected advanced fibrosis, as it provides a non-invasive estimate of liver stiffness [122]. According to the 2024 EASL guidelines, elastography is emphasized for fibrosis staging rather than relying on ultrasound alone. In select cases, liver biopsy is indicated when non-invasive tests yield conflicting results or if there is clinical suspicion of advanced fibrosis, cirrhosis, or overlapping liver diseases. In the diagnostic work-up for MASLD, additional tests are necessary to exclude other liver conditions that may coexist or mimic MASLD, including hepatitis B and C panels, autoimmune liver disease markers (ANA and ASMA), and iron studies to rule out hereditary hemochromatosis. This comprehensive approach, combining metabolic risk assessment, non-invasive fibrosis evaluation, imaging, and laboratory testing, ensures an accurate diagnosis and appropriate risk stratification for patients with suspected MASLD.

## 4. Approaches to Managing MASLD and Halting Its Progression

Intervening in MASLD during its early stages, when hepatic steatosis is the primary finding, is considered the most effective approach to managing the condition [120,123]. Early interventions not only increase the likelihood of preventing long-term complications but also contribute to improving overall patient health outcomes [124,125].

Lifestyle modifications offer additional benefits by addressing other components of metabolic syndrome, such as dyslipidemia, hypertension, and insulin resistance. These improvements help reduce the risk of cardiovascular disease, a major comorbidity in MASLD patients [126]. Among these interventions, achieving a weight loss of 7–10% is particularly recommended to attain these metabolic and cardiovascular benefits [12].

Dietary modifications play a crucial role in managing MASLD. Reducing the consumption of harmful foods and adopting a healthy dietary pattern, such as the Mediterranean diet, has been shown to provide significant benefits. This diet, characterized by its richness in fruits, vegetables, whole grains, and unsaturated fats, helps improve hepatic steatosis, reduce systemic inflammation, and slow the progression of the disease [12,120,123,126,127].

Weight loss, achieved through a combination of dietary modifications, physical activity, or pharmacological treatments such as semaglutide, has demonstrated profound impacts on MASLD management. These interventions not only improve liver-related outcomes but also positively affect the pathophysiology of type 2 diabetes, as well as associated cardiovascular and renal complications, thereby addressing multiple comorbidities simultaneously [110].

Additionally, Vitamin E, a potent antioxidant, has shown promise in clinical trials by reducing hepatic steatosis and inflammation, two key drivers of MASLD progression. This effect has been observed primarily in non-diabetic patients [12,112]. However, further research is required to fully understand its potential long-term risks, particularly in diverse patient populations [12].

Coffee is a rich source of bioactive compounds, including caffeine, diterpenoid alcohols, potassium, niacin, and chlorogenic acid, which collectively exhibit antioxidant and antifibrotic properties. These components contribute to the protective effects of coffee against liver-related metabolic and fibrotic conditions. For instance, caffeine, a well-studied coffee component, has been proposed to prevent or reverse liver fibrosis by acting as an adenosine A2A receptor antagonist, thereby inhibiting the activation of hepatic stellate cells—a key driver of fibrogenesis. Additionally, chlorogenic acid and other polyphenols in coffee contribute to its antioxidant properties, which help mitigate oxidative stress, a critical factor in the progression of liver diseases [128].

One of the lesser-known but significant bioactive compounds in coffee is N-methylpyridinium (NMP), an alkaloid found in coffee brew. At low concentrations (0.1–0.25 µM, equivalent to the amount in one to three espresso coffees), NMP has been shown to inhibit lipid accumulation in hepatic cells and reduce ROS levels. This effect is closely linked to the modulation of endoplasmic reticulum stress, a key player in lipid metabolism and oxidative stress. Endoplasmic reticulum stress activation promotes the upregulation of sterol regulatory element-binding protein 1 (SREBP-1) and the expression of lipogenic genes, leading to increased lipid accumulation. Furthermore, endoplasmic reticulum stress induces ROS production in both the endoplasmic reticulum and mitochondria, exacerbating oxidative damage. NMP counteracts these effects by reducing endoplasmic reticulum stress, as evidenced by the downregulation of endoplasmic reticulum stress markers such as XBP1, ATF6, CHOP, GRP78, and P-eIF2α. By alleviating this stress, NMP restores normal levels of SREBP-1 and its lipogenic gene targets, thereby reducing lipid accumulation and diminishing ROS production, highlighting its antioxidant properties [129].

The protective effects of coffee consumption on liver health are further supported by epidemiological evidence. A meta-analysis of seven studies involving 4825 coffee consumers and 49,616 non-consumers evaluated the relationship between coffee intake—ranging from 0 to over 5 cups per day—and metabolic-dysfunction-associated steatotic liver disease (MASLD). The findings revealed that consuming more than three cups of coffee daily is associated with a significantly reduced risk of developing MASLD, as well as liver fibrosis and cirrhosis [130]. These findings align with current international guidelines, which recommend a daily intake of at least three cups of coffee, whether caffeinated or decaffeinated, for individuals without contraindications [116].

On the other hand, bariatric surgery is a well-established treatment for obesity that not only promotes significant weight loss but also improves metabolic syndrome and reduces hepatic fat accumulation, inflammation, and fibrosis [131]. This surgical intervention has demonstrated remarkable efficacy in addressing MASLD, particularly in individuals with morbid obesity or a high BMI accompanied by metabolic-syndrome-related comorbidities. A retrospective study evaluating the long-term effects of bariatric surgery found that, five years post-surgery, 84% of patients (95% CI: 73.1–92.2%) achieved the resolution of steatohepatitis without worsening fibrosis, while fibrosis itself decreased by 70.2% (95% CI: 56.6–81.6%) [132].

Among the various bariatric procedures, the Roux-en-Y gastric bypass (RYGB) has shown particular promise in improving fibrotic MASH. Specifically, RYGB resulted in a marked reduction in the fibrotic NASH index (FNI) score, with 64% of patients diagnosed with fibrotic MASH at baseline demonstrating significant improvement post-surgery. These results align with previous findings by Pais et al., who reported that patients with high histological activity grades experienced substantial improvements following bariatric surgery, including an 80% resolution rate of MASH [133].

Together, these findings underscore the potential of RYGB as an effective intervention for improving fibrotic MASH, particularly in patients with advanced disease [134].

The consistent evidence from these studies highlights the role of bariatric surgery in addressing not only metabolic and weight-related outcomes but also liver-specific pathologies. By reducing hepatic fat, inflammation, and fibrosis, bariatric surgery offers a promising therapeutic approach for patients with MASLD and fibrotic MASH, particularly those with advanced disease and obesity-related metabolic complications. These outcomes emphasize the importance of considering bariatric surgery as a viable treatment option for individuals with severe liver disease and obesity, providing both metabolic and hepatic benefits.

### 4.1. Drugs in Development and Research for MASLD and MASH Treatmen

#### 4.1.1. Resmetirom

Resmetirom is the first, and, currently, the only, FDA-approved medication for the treatment of MASH with moderate to advanced fibrosis. Its approval was granted in March 2024, marking a significant milestone in the management of this condition [135,136,137].

This drug functions as a selective agonist of the thyroid hormone receptor beta (THR-β), specifically targeting the liver. In MASH, the role of THR-β is impaired, leading to reduced mitochondrial function, diminished fatty acid β-oxidation, and an increase in fibrosis. Resmetirom directly addresses these issues, restoring some of the lost metabolic activity [137,138]. By reducing lipotoxicity and enhancing mitochondrial function in hepatocytes, Resmetirom likely contributes to its observed anti-inflammatory and antifibrotic effects. Additionally, the medication has demonstrated a positive impact on lipid profiles, particularly by lowering low-density lipoprotein (LDL) cholesterol levels. These combined benefits highlight its potential for comprehensive metabolic improvement in patients with MASH [137,139].

Nevertheless, like any medication, Resmetirom is associated with some side effects. The most reported adverse effects include diarrhea and nausea, which are typically mild and self-limiting, and occur early in the course of treatment. Importantly, clinical trials have shown that the drug does not negatively impact heart rate or body weight, which further supports its safety profile [136,138]. Although Resmetirom has already received FDA approval, ongoing studies are focused on evaluating its long-term effects and determining its impact on clinical outcomes. These investigations aim to provide further insights into its efficacy and safety over extended periods of use [138,139].

Other drugs have not yet been approved by the FDA and are still under investigation, which we will discuss later. We will also outline their potential benefits in MASLD, as well as their adverse effects (Table 2).

#### 4.1.2. PPAR

PPARs are a group of nuclear receptors that play an essential role in regulating lipid and glucose metabolism. Due to this critical function, they are considered attractive therapeutic targets for MASLD and MASH. Several drugs have been developed to target these receptors. While some act selectively on the α or γ subtypes, others exhibit dual activity or act more broadly as agonists across all PPAR subtypes.

One notable drug is Pemafibrate, a selective PPAR-α modulator. Pemafibrate has demonstrated a significant ability to improve lipid parameters and reduce ALT levels without causing an increase in creatinine, which is a common concern with other treatments [140].

On the other hand, Saroglitazar targets both the α and γ isoforms, making it a dual PPAR-α/γ agonist. This drug has proven effective in treating atherogenic dyslipidemia, which often accompanies MASLD. Its mechanism of action includes promoting fatty acid oxidation, reducing VLDL production, and decreasing apolipoprotein C-III levels. Additionally, Saroglitazar induces the expression of PPAR-γ-sensitive genes involved in carbohydrate and lipid metabolism. These effects collectively reduce the hepatic metabolic load and improve glycemic control [141].

Lastly, Pioglitazone, a PPAR-γ agonist, is a well-known hypoglycemic agent primarily used to treat T2D. However, it has also shown remarkable benefits in MASLD and MASH [142].

Clinical trials with Pioglitazone have demonstrated its capacity to reduce hepatic fat content while acting as an insulin sensitizer. Improving insulin sensitivity helps regulate blood glucose levels, a critical factor in managing MASLD. Furthermore, Pioglitazone exerts additional metabolic benefits, including reductions in triglycerides, LDL cholesterol, and free fatty acids. Due to these multifaceted effects, Pioglitazone is considered unique among antihyperglycemic agents for its ability to improve liver histology in patients with MASH [142].

The development of drugs targeting PPARs offers a promising avenue for addressing the complex metabolic and histological changes associated with MASLD and MASH. Each drug exhibits distinct mechanisms and benefits, providing a tailored approach to managing these challenging conditions.

#### 4.1.3. SGLT2

Another class of drugs is the sodium-glucose cotransporter (SGLT) inhibitors, which have demonstrated numerous beneficial effects, particularly in MASLD and MASH. In addition to their primary use in treating MASLD, these drugs hold promising potential for addressing multiple metabolic and hepatic abnormalities [143].

Selective SGLT2 inhibitors primarily act on the kidneys by inhibiting glucose reabsorption in the renal tubules. This action promotes glucose excretion through the urine, which subsequently lowers blood glucose levels and improves insulin resistance [144]. Furthermore, the excretion of glucose leads to weight loss, a critical therapeutic goal in MASLD management [144,145]. In addition to selective SGLT2 inhibitors, some drugs target both SGLT1, which blocks intestinal glucose absorption, and SGLT2 in the kidneys. This dual mechanism of action provides additional metabolic and hepatic benefits [146].

In addition to these effects, several studies have shown that SGLT2 inhibitors significantly reduce hepatic steatosis [143,144,146]. Interestingly, this reduction has been observed not only in patients with T2D but also in those without the condition [146]. Moreover, improvements have been consistently reported in serum markers of liver injury, including ALT, AST, and GGT, further supporting the liver-protective effects of these drugs [144,146]. Another noteworthy benefit of SGLT2 inhibitors lies in their positive impact on lipid profiles. These drugs have been associated with reductions in triglyceride levels and increases in HDL cholesterol, which contribute to an improved cardiovascular risk profile [147].

However, the effects of SGLT2 inhibitors on hepatic fibrosis remain a subject of debate. While some studies suggest that these drugs may improve hepatic fibrosis in patients with MASH—as evidenced by reductions in liver stiffness measured by transient elastography—other studies show a different result [143,146].

Examples of SGLT2 inhibitors that have been recognized as potential therapeutic options for MASLD and MASH due to their wide-ranging benefits include dapagliflozin, empagliflozin, ipragliflozin, canagliflozin, and ertugliflozin [143,146,148,149].

Licogliflozin is a dual SGLT1 and SGLT2 inhibitor. This drug has shown significant reductions in hepatic steatosis and improvements in liver enzyme levels, such as ALT, AST, and GGT. Furthermore, it has been associated with improvements in hepatic fibrosis markers. Although licogliflozin has not demonstrated significant changes in plasma lipid levels or insulin resistance, it has been shown to reduce HbA1c levels, indicating its potential utility in glycemic control [146].

#### 4.1.4. FXR Agonist

Several drugs have been developed to target the farnesoid X receptor (FXR), either as standalone therapies or in combination with other treatments. FXR is a nuclear receptor that plays a crucial role in bile acid homeostasis, lipid and glucose metabolism, oxidative stress, and inflammation [150,151]. By targeting FXR, these drugs aim to modulate metabolic pathways and address the underlying mechanisms of liver disease. FXR agonists are drugs that bind to and activate FXR, triggering a cascade of metabolic effects that contribute to improved liver function and reduced disease progression.

Second-generation FXR agonists, such as Vonafexor, have been designed to activate FXR in a more potent and selective manner. Studies have demonstrated that Vonafexor effectively reduces hepatic steatosis and improves markers of liver injury. Furthermore, it contributes to the management of MASLD by promoting weight loss and improving renal function. Dosing studies for Vonafexor have revealed that doses of 100 mg and 200 mg per day result in significant reductions in hepatic steatosis. In contrast, higher doses, such as 400 mg, do not provide additional efficacy benefits and are associated with an increased incidence of side effects. These side effects, which are like those observed with other FXR agonists, include mild to moderate pruritus and alterations in lipid profiles [152].

Another promising FXR agonist under development is EDP-305. This drug has shown beneficial effects similar to OCA, including reductions in C4 levels and increased fibroblast growth factor 19 (FGF19) levels. However, EDP-305 has the added advantage of causing less significant increases in LDL cholesterol compared to OCA and other drugs in this class, such as Aldafermin. This distinction may make it a more favorable option for certain patient populations [152].

Cilofexor, another FXR agonist, operates through a mechanism of action similar to other drugs in this class. While it shares many of the beneficial effects of FXR agonists, it has been found to be less potent in reducing hepatic steatosis and ALT levels [152]. Nevertheless, it remains a potential therapeutic option due to its overall tolerability and safety profile.

By targeting FXR, these drugs offer a range of benefits, including improved liver histology, metabolic regulation, and reduced liver-related inflammation. However, careful consideration of dosing and side effects is essential in order to optimize their therapeutic potential.

#### 4.1.5. Obeticholic Acid

Obeticholic acid (OCA) has been extensively studied in patients with MASH and cirrhosis. It has been shown to reduce bile acid synthesis by lowering C4 levels (a precursor of bile acids) and increasing the levels of FGF19. Additionally, OCA exhibits significant anti-inflammatory and anti-fibrotic effects [153].

Clinical studies have consistently demonstrated that OCA improves liver histology by reducing fibrosis, inflammation, and cellular damage [150,154]. However, its use is associated with some common but reversible side effects, including pruritus, increased bile lithogenicity, and elevated LDL cholesterol levels.

Obeticholic acid, an FXR agonist, has shown encouraging results by improving liver fibrosis and enzyme levels. However, its side effects, such as pruritus and elevated LDL cholesterol, have sparked discussions about its risk–benefit profile [12].

#### 4.1.6. GLP-1 Receptor Agonist

Glucagon-like peptide-1 (GLP-1) receptors are the target of several drugs, either as direct agonists or in combination with other medications. GLP-1, a hormone primarily secreted by L cells in the intestines, functions as an incretin. This means it promotes insulin secretion in response to glucose, thereby maintaining glucose levels under control through proportional insulin excretion. Importantly, GLP-1 receptors are widely distributed throughout the body, which accounts for the hormone’s various non-glycemic effects. In addition to its primary role in glucose regulation, GLP-1 receptor agonists offer numerous benefits, particularly for patients with MASLD. A key advantage of these drugs is their ability to regulate appetite and induce satiety, which, in turn, contributes to weight reduction. This effect is mediated through the modulation of gastric emptying, delaying the rate at which food moves from the stomach to the small intestine. Moreover, GLP-1 receptor agonists positively influence lipid metabolism, blood pressure, and cardiovascular health, while also exhibiting potential benefits for cognitive function and mood [155]. These broad effects make these drugs highly effective for managing T2D and obesity, where they have long been used as first-line treatments.

Recently, however, research has expanded their potential applications to MASLD, with promising results in reducing hepatic steatosis and improving overall metabolic function [155]. Some drugs currently under investigation for the treatment of MASLD and MASH include semaglutide, dulaglutide, liraglutide, exenatide, cotadutide, and beinaglutide [135,155,156,157,158,159,160].

GLP-1 receptor agonists have demonstrated significant reductions in hepatic steatosis, largely attributed to improvements in insulin sensitivity and decreased de novo lipogenesis [156,161]. Additionally, these drugs exhibit anti-inflammatory properties, which are critical for mitigating hepatic inflammation—a key aspect of MASH pathophysiology. For instance, liraglutide has been shown to modulate the TLR4/NF-κB inflammatory pathway, thereby reducing liver injury [156].

Interestingly, although GLP-1 receptors are not directly expressed in the liver, the beneficial effects of GLP-1 receptor agonists on hepatic function are thought to be primarily indirect. These include weight loss, improved glycemic control, and enhanced insulin sensitivity. Nonetheless, emerging studies suggest that these drugs may also have direct effects on hepatocytes by reducing lipogenesis and increasing fatty acid oxidation [156,161].

Further advancing this therapeutic area, dual agonists that target both GLP-1 and glucose-dependent insulinotropic peptide receptors, such as tirzepatide, have shown enhanced efficacy in controlling glucose levels and body weight compared to traditional GLP-1 receptor agonists. This dual mechanism of action allows for superior metabolic outcomes [162,163]. Additionally, dual agonists such as pemvidutide and survodutide, which target both GLP-1 and glucagon receptors, have demonstrated substantial reductions in hepatic steatosis, liver stiffness, and biomarkers of liver fibrosis [163,164].

The development of triple receptor agonists, which simultaneously target GLP-1, glucose-dependent insulinotropic peptide, and glucagon receptors, represents a further innovation in this field. Notable examples include retatrutide and efocipegtrutide. Retatrutide has shown the potential to reduce hepatic steatosis, accompanied by changes in adiponectin, leptin, triglycerides, and FGF21 levels—factors closely associated with lipid metabolism and insulin sensitivity. Similarly, efocipegtrutide has demonstrated reductions in hepatic steatosis, though it is still in phase 2 trials for the treatment of MASH [163,165].

#### 4.1.7. DPP-4

Dipeptidyl peptidase-4 (DPP-4) is an enzyme responsible for the degradation of incretin hormones, including GLP-1 and the glucose-dependent insulinotropic peptide. The inhibition of this enzyme results in increased levels of incretins, which, in turn, enhances insulin secretion and suppresses glucagon release in the pancreas, thereby contributing to improved glycemic control [166]. Notably, DPP-4 is widely distributed in the liver and is overexpressed in MASLD. Moreover, serum levels of DPP-4 have been shown to correlate with the severity of hepatic steatosis, suggesting a direct link between DPP-4 activity, liver damage, and hepatic lipogenesis [142,166].

Given the enzyme’s significant role in the pathophysiology of MASLD, DPP-4 inhibition represents a promising therapeutic strategy to prevent or delay the progression of the disease. This therapeutic potential is further supported by evidence demonstrating that DPP-4 inhibitors can modulate inflammatory pathways and fibrosis. Specifically, these drugs reduce the expression of pro-inflammatory mediators, attenuate endoplasmic reticulum stress, decrease hepatocyte apoptosis, and lower the accumulation of fibronectin and alpha-actin, key markers of fibrosis [166].

Several drugs within this class, such as sitagliptin, vildagliptin, evogliptin, and saxagliptin, are commonly used in the treatment of T2D. However, recent studies have also explored their potential benefits in MASLD and MASH. These investigations have revealed that DPP-4 inhibitors can modulate fatty acid metabolism, reduce de novo lipogenesis, and improve hepatic glucose metabolism, thereby offering potential therapeutic advantages for these liver conditions [147,166,167].

#### 4.1.8. FGF Analogs

Fibroblast growth factor (FGF) 21 is a hormone that plays a pivotal role in glucose and lipid metabolism, as well as in maintaining energy homeostasis. Additionally, it regulates the secretion of adiponectin, an adipokine with multiple beneficial effects, including insulin-sensitizing, anti-steatotic, anti-inflammatory, and anti-fibrotic properties [168,169,170]. FGF-21 exerts its effects by enhancing the mitochondrial capacity and activating the antioxidant pathways, thereby protecting cells from oxidative stress. Furthermore, it helps prevent hallmark features of MASH, such as hepatocyte death, inflammation, and fibrosis [170].

Despite these promising effects, the clinical utility of endogenous FGF-21 is limited due to its short half-life, which restricts its therapeutic potential. To address this limitation, researchers have developed long-acting FGF-21 analogs, including Pegbelfermin, Pegozafermin, and Efruxifermin [168]. Although these analogs share similar mechanisms of action, they differ in their structural properties and pharmacokinetics. Pegbelfermin, for instance, is conjugated with polyethylene glycol to extend its half-life, allowing for weekly dosing. Similarly, Pegozafermin is a glycopegylated, long-acting analog that enables dosing intervals of up to two weeks. Meanwhile, Efruxifermin is composed of two covalently bound FGF-21 chains, which confer a higher affinity for its receptors and enhance its therapeutic efficacy [146,168,169].

In addition to FGF-21, FGF-19 has garnered attention for its distinct metabolic functions. FGF-19 primarily regulates bile acid metabolism but also offers benefits beyond glucose homeostasis. Specifically, it reduces bile acid synthesis in the liver by suppressing cholesterol 7α-hydroxylase (CYP7A1), a key enzyme in the bile acid synthesis pathway. Moreover, FGF-19 may exert additional metabolic effects by modulating the intestinal microbiome. Therapeutically, Aldafermin, an FGF-19 analog, has been developed to harness these properties. Aldafermin has shown potential not only in regulating bile acid metabolism but also in improving liver health, particularly in conditions such as MASH. FGF-19 is also considered a marker of FXR activation, highlighting its potential as a therapeutic target for MASH treatment [171].

#### 4.1.9. Statins

Rosuvastatin and atorvastatin belong to a group of medications known as statins, widely utilized for cardiovascular risk reduction and improving the lipid profile in patients with or without liver diseases, such as MASLD or MASH [172]. Beyond their cardiovascular benefits, research has demonstrated their ability to reduce lipid accumulation in the liver. For example, a study found that rosuvastatin significantly decreased intrahepatic lipids by 42.3%, highlighting its potential role in addressing hepatic steatosis [173]. In certain cases, statins are used in combination with other therapeutic agents, such as FGF-19 analogs, to achieve better lipid profile optimization and enhance patient outcomes [171].

There are numerous other statins available in clinical practice, including simvastatin, pravastatin, lovastatin, fluvastatin, and pitavastatin. While all these statins function by inhibiting HMG-CoA reductase, there are variations in their potency, as well as differences in their adverse effects and potential drug interactions. These differences make it beneficial to study their efficacy and safety in the context of MASLD or MASH [174].

#### 4.1.10. Metformin

Metformin, a first-line treatment for T2D, exerts its effects primarily by enhancing insulin sensitivity in both peripheral tissues and the liver. Beyond its glucose-lowering properties, metformin has been shown to inhibit hepatic gluconeogenesis and improve fatty acid metabolism, thus providing metabolic benefits for the liver. A meta-analysis revealed that metformin could reduce ALT and AST levels in patients with MASLD, in addition to lowering triglyceride and total cholesterol levels. Moreover, metformin improves insulin resistance, which is a central mechanism underlying MASLD pathogenesis. Despite these promising effects, the clinical efficacy of metformin in MASLD and MASH remains a topic of debate. While several studies support its therapeutic benefits, others report limited or no significant impact on the progression of these conditions [175].

#### 4.1.11. Promising New Medications

In addition to statins and metformin, several emerging therapies, some of which are still under investigation, show promise as potential treatments for MASLD or MASH.

One such agent is Firsocostat, an inhibitor of acetyl-CoA carboxylase (ACC), a key enzyme involved in de novo lipogenesis. By inhibiting ACC, Firsocostat reduces hepatic lipid production, thereby decreasing hepatic steatosis. Clinical studies have demonstrated that Firsocostat is well-tolerated and effectively reduces hepatic steatosis in patients with MASH. Moreover, its combination with other drugs, such as semaglutide and cilofexor, has shown synergistic benefits, as these agents target different pathways in MASH pathogenesis. Notably, the combination of semaglutide and Firsocostat has resulted in greater improvements in hepatic steatosis and liver biochemistry compared to semaglutide alone [153].

Aramchol, another investigational drug, partially inhibits the expression of stearoyl-CoA desaturase 1 (SCD1), an enzyme involved in fatty acid synthesis. In preclinical models, Aramchol has been shown to reduce liver triglyceride content and fibrosis. Phase 2 clinical trials have further demonstrated significant reductions in hepatic steatosis, as assessed by magnetic resonance spectroscopy [176].

Selonsertib, an inhibitor of apoptosis signal-regulating kinase 1 (ASK1), targets pathways involved in inflammation and apoptosis, two critical aspects of MASLD pathophysiology. While Selonsertib effectively inhibited its target, clinical trials failed to demonstrate the regression of fibrosis or the halting of disease progression in MASH patients. However, the drug showed improvements in the enhanced liver fibrosis score and liver stiffness, as assessed by transient elastography [177].

Finally, PF-06835919, one of the most recent drugs under investigation, has demonstrated the ability to reduce hepatic steatosis in patients with MASLD and T2D. In addition to its effects on liver fat, PF-06835919 has shown improvements in glucose levels, insulin sensitivity, and the homeostatic model assessment of insulin resistance. Although further studies are required, this drug appears to hold significant promise for treating MASLD and related conditions [178].

The landscape of MASLD/MASH treatment is rapidly evolving, with multiple promising therapies under investigation. Future research focusing on combination treatments, biomarker discovery, personalized approaches, and lifestyle modifications will be crucial in effectively managing this complex disease (Table 3).

**Table 2 ijms-26-02959-t002:** Experimental treatments in MASLD. ↑ increase, ↓ decrease.

Target	Action	Drugs Name	Benefits	Side Effects	References
THRβ	Agonist	Resmetirom *	↓ MASH, ↓ Fibrosis, ↓ LDL, ↓ TG, ↓ ALT, ↓ AST, ↓ GGT	Diarrhea, nausea, pruritus, vomit	[138]
PPAR	PPARα agonist	Pemafibrate	↓ Liver stiffness, ↓ ALT, ↓ TG, ↑HDL	Diarrhea, nausea, abdominal pain, CKD, AKI, skin rashes, muscle pain	[140]
		Saroglitazar	↓ TC, ↓ TG, ↓ LDL, ↓ VLDL	Diarrhea, cough, abdominal pain, fatigue, nausea, dyspepsia	[141]
		Fenofibrate	↓ TG, ↓ LDL, ↑HDL, ↓ ALP, ↓ GGT, ↓ Insulin resistance	Nausea, vomit, diarrhea, abdominal pain, headache, dizziness, rashes	[140]
	PPARγ agonist	Pioglitazone	↓ MASH, ↓ Fibrosis, ↓ ALT, ↓ AST, ↓ HbA1c, ↓ FPG, ↓ Insulin resistance, ↓ TG, ↓ LDL	Weight gain, fluid retention, nausea, vomit, lethargy, insomnia	[145]
	PanPPAR agonist	Lanifibranor	↓ MASH, ↓ Fibrosis, ↓ TG, ↑HDL, ↓ Insulin resistance, ↓ HbA1c, ↓ FPG	Nausea, diarrhea, anemia, peripheral edema, weight gain	[141]
SGLT2	Inhibitor	Canagliflozin	↓ Steatosis, ↓ Fibrosis, ↓ TG, ↑Insulin Sensitivity, ↑Insulin secretion, ↓ Body and fat mass, ↓ Liver enzymes, ↓ Visceral fat	Nausea, diarrhea, constipation, UTIs, genital mycotic infections, increased urination, increased thirst	[148]
		Dapagliflozin	↓ Steatosis, ↓ Pancreatic fat content, ↓ Inflammatory cytokine ↓ ALT, ↓ Visceral fat ↓ Body weight, ↑Insulin sensitivity	UTIs, genital mycotic infections, increased urination, episodes of hypotension, upper respiratory tract infections, headache, rash	[144]
		Empagliflozin	↓ Steatosis, ↓ Fibrosis, ↓ Visceral fat, ↓ AST, ↓ ALT, ↑Insulin sensitivity, ↓ Body weight	UTIs, genital mycotic infections, increased urination	[179]
		Ipragliflozin	↓ Steatosis, ↓ Fibrosis, ↓ Visceral fat, ↑HDL, ↓ TG, ↓ FPG, ↓ HbA1c, ↓ Blood pressure, ↓ Liver enzymes, ↓ Uric acid	UTIs, hypoglycemia, increase urinary glucose excretion, genital mycotic infections	[143]
		Licogliflozin	↓ Steatosis ↓ Fibrosis, ↓ ALT, ↓ AST, ↓ GGT, ↓ Body weight, ↓ HbA1c, ↑eGFR	Diarrhea, flatulence, abdominal distension, renal impairment, UTIs, genital mycotic infections	[146]
		Ertugliflozin	↓ Steatosis, ↓ ALT, ↓ AST, ↓ GGT, ↓ TG, ↓ Body weight, ↑Insulin sensitivity, ↓ Uric acid	Genital mycotic infections, UTIs, headache, back pain	[149]
FXR	Agonist	EDP-305	↓ Steatosis, ↓ Fibrosis, ↓ ALT, ↓ GGT, ↓ TG, ↓ FPG	Pruritus, diarrhea, abdominal discomfort, gastroesophageal reflux, headache, dizziness, vomit	[150]
		Obeticholic acid	↓ Steatosis, ↓ Fibrosis, ↑Insulin sensitivity, ↓ Body weight, ↓ ALT, ↓ AST, ↓ GGT	Pruritus, fatigue, abdominal pain and discomfort, increases in LDL-C, dizziness, constipation, arthralgia, eczema, thyroid function abnormality	[154]
		Vonafexor	↓ Steatosis, ↓ MASH, ↓ Fibrosis, ↓ Body weight, ↓ ALT, ↓ GGT, ↑eGFR	Pruritus, increased LDL-C, dyslipidemia, nausea, fatigue	[152]
		Cilofexor	↓ Steatosis, ↓ Fibrosis, ↓ ALT, ↓ AST	Pruritus, dyslipidemia, nausea, diarrhea, vomit, constipation and decreased appetite, increased LDL-C, increases TG	[153]
FGF	19 analogue	Aldafermin	↓ MASH, ↓ Fibrosis, ↓ ALT, ↓ AST, ↑Insulin sensitivity	Diarrhea, abdominal pain, nausea, altered bowel function, increased appetite, increased LDL-C, fatigue, headache, constipation	[171]
	21 analogue	Pegbelfermin	↓ Steatosis, ↓ Fibrosis, ↓ ALT, ↓ AST	Diarrhea, nausea, increased appetite, local reactions, headache, fatigue, hypoglycemia	[168]
		Pegozafermin	↓ Steatosis, ↓ Fibrosis, ↓ TG, ↑HDL, ↓ LDL, ↓ ALT, ↓ HbA1c, ↑Insulin sensitivity	Diarrhea, nausea, vomit, increased appetite, local reactions	[169]
		Efruxifermin	↓ Steatosis, ↓ MASH, ↓ Fibrosis, ↓ ALT, ↓ AST, ↓ TG, ↓ LDL, ↑HDL, ↑Insulin sensitivity, ↓ HbA1c, ↓ Uric acid	Diarrhea, nausea, increased appetite, local reactions	[170]
DPP4	Inhibitor	Sitagliptin	↓ Steatosis, ↓ Liver stiffness, ↓ HbA1c, ↓ FPG, ↓ TG, ↓ LDL, ↑Insulin sensitivity, ↓ Atherosclerosis	Upper respiratory tract infections, headache, diarrhea, UTIs, joint pain	[147]
		Vildagliptin	↓ Steatosis, ↓ Liver stiffness, ↓ HbA1c, ↓ FPG, ↓ Lipid profile, ↓ ALT, ↓ Atherosclerosis	Upper respiratory tract infections, headache, diarrhea, nausea, dizziness	[166]
		Evogliptin	↓ Steatosis, ↓ HbA1c, ↓ FPG, ↑ Insulin sensitivity, ↓ AST, ↓ ALT	Hypoglycemia, nasopharyngitis, headache, gastrointestinal disturbance	[142]
		Saxagliptin	↓ Steatosis, ↓ HbA1c, ↓ FPG, ↓ Liver enzymes, ↓ Adipose tissue	Upper respiratory tract infections, UTIs, diarrhea, vomit, abdominal pain, headache, hypoglycemia, lymphopenia	[167]
HMG-CoA	Inhibitor	Atorvastatin	↓ Steatosis, ↓ MASH, ↓ LDL, ↑ HDL, ↓ TG, ↑Insulin sensitivity ↓ CRP, ↓ VLDL	Myalgia, myopathy, constipation, diarrhea, dyspepsia, flatulence, abdominal pain, nausea, headache, insomnia, drowsiness	[172]
		Rosuvastatin	↓ Steatosis, ↓ LDL, ↓ TG, ↑HDL, ↓ VLDL	Muscle toxicity, dizziness, headache, myalgia, gastrointestinal issues	[173]
AMPK	Agonist	Metformin	↓ Steatosis, ↑Insulin sensitivity, ↓ TG, ↓ Body weight, ↓ ALT, ↓ AST,	Nausea, diarrhea, abdominal pain, bloating, flatulence, vomit, metallic taste	[175]
GLP-1	Agonist	Semaglutide	↓ Steatosis, ↓ MASH, ↓ Body weight, ↓ HbA1c, ↓ ALT, ↓ AST, ↓ CRP, ↓ TG, ↓ LDL, ↓ VLDL	Nausea, vomit, diarrhea, constipation, abdominal pain, headache, fatigue, dyspepsia, dizziness, hypoglycemia	[135]
		Dulaglutide	↓ Steatosis, ↓ HbA1c, ↑Insulin sensitivity, ↓ ALT, ↓ AST	Nausea, vomit, diarrhea, abdominal pain, dyspepsia, decreased appetite, headache	[155]
		Liraglutide	↓ Steatosis, ↓ Fibrosis, ↓ ALT, ↓ AST, ↓ Body weight, ↓ TG, ↓ HbA1c, ↑Insulin sensitivity	Nausea, vomit, diarrhea, constipation, dyspepsia, abdominal pain	[156]
		Exenatide	↓ Steatosis, ↓ Liver enzymes, ↓ Body weight, ↑Insulin sensitivity	Nausea, vomit, diarrhea, hypoglycemia, local reactions, upper respiratory symptoms	[157]
		Cotadutide	↓ Steatosis, ↓ Fibrosis, ↓ Body weight, ↓ HbA1c	Nausea, vomit, diarrhea, constipation, decreased appetite, hypoglycemia	[158]
		Beinaglutide	↓ Steatosis, ↓ Fibrosis, ↓ Body weight, ↓ FPG, ↓ HbA1c	Diarrhea, nausea, vomit, dizziness	[159]
GLP-1/GIP	Dual agonist	Tirzepatide	↓ Steatosis, ↓ MASH, ↓ Fibrosis, ↓ FPG, ↓ HbA1c, ↓ Body weight, ↓ TG, ↑HDL, ↑Insulin sensitivity, ↓ OSA	Nausea, diarrhea, vomit, abdominal pain, constipation, decreased appetite, dyspepsia	[162]
GLP-1/Glucagon	Dual agonist	Survodutide	↓ Steatosis, ↓ Liver stiffness, ↓ Body weight, ↓ HbA1c	Nausea, vomit, diarrhea, dehydration, angioedema	[164]
		Pemvidutide	↓ Steatosis, ↓ ALT, ↓ Body weight, ↓ Blood pressure, ↓ Liver volume, ↓ Lipid profile	Nausea, vomit, diarrhea, constipation	[161]
GLP-1/Glucagon/GIP	Triple agonist	Retatrutide	↓ Steatosis, ↓ Body weight, ↓ Adipose tissue	Nausea, diarrhea, vomit, constipation, change in bowel habits, skin hyperesthesia	[163]
		Efocipegtrutide	↓ MASH	Hyperglycemia, gastrointestinal issues	[165]
ACC	Inhibitor	Firsocostat	↓ Steatosis, ↓ ALT, ↓ Body weight, ↓ Blood pressure, ↓ Liver volume, ↓ Lipid profile	Hypertriglyceridemia, hypercholesterolemia, nauseas, diarrhea, pruritus	[153]
PDE	Inhibitor	Pentoxifylline	↓ Steatosis, ↓ MASH, ↓ ALT, ↓ AST	Nausea, vomit, diarrhea, anorexia, dizziness, headache, chest pain, tachycardia, skin reactions	[180]
SCD1	Partial inhibitor	Aramchol	↓ Steatosis, ↓ MASH, ↓ Fibrosis, ↓ ALT, ↓ AST, ↓ HbA1c, ↑Insulin sensitivity	Skin disorders, eye disorders, atherosclerosis, hair loss, hypothermia	[176]
ASK1	Inhibitor	Selonsertib	↓ Liver stiffness, ↓ MASH	Headache, nausea, sinusitis, nasopharyngitis, upper abdominal pain, back pain, fatigue	[177]
Ketohexokinase	Inhibitor	PF-06835919	↓ Steatosis, ↑Insulin sensitivity, ↓ FPG	Headache, nausea, insomnia, back pain, dyspepsia, fatigue	[178]

Abbreviations: THRβ, Thyroid hormone receptor-beta; PPAR, Peroxisome proliferator-activated receptor; SGLT2, Sodium-Glucose Transport Protein 2; FXR, Farnesoid X receptor; FGF, Fibroblast growth factor; DPP4, Dipeptidyl peptidase 4; HMG-CoA, Hydroxymethylglutaryl coenzyme A; AMPK, AMP-activated protein kinase; GLP-1, Glucagon-like peptide 1; GIP, Gastric inhibitory polypeptide; ACC, Acetyl-Coenzyme A carboxylase; PDE, Phosphodiesterase; SCD1, Stearoyl-Coenzyme A desaturase 1; ASK1, Apoptosis signal-regulating kinase 1; MASH, Metabolic-dysfunction-associated steatohepatitis; LDL, Low-density lipoprotein; VLDL, Very-low-density lipoprotein; HDL, High-density lipoprotein; TG, Triglycerides; ALT, Alanine transaminase; AST, Aspartate aminotransferase; GGT, Gamma-glutamyl transferase; ALP, Alkaline phosphatase; FPG, Fasting plasma glucose; eGFR, Estimated glomerular filtration rate; HbA1c, Hemoglobin A1C; CRP, C-reactive protein; OSA, Obstructive sleep apnea; CKD, chronic kidney disease; AKI, acute kidney injury; UTIs, urinary tract infections. * Approved by the FDA.

It is equally important to manage comorbidities contributing to MASLD, such as diabetes, hypertension, and dyslipidemia, through an integrated and patient-centered approach. A multidisciplinary team, comprising healthcare professionals such as nutritionists, psychologists, and specialists from various medical fields, is essential in order to deliver comprehensive care tailored to each patient’s unique needs. Furthermore, future strategies for the safe and effective management of MASLD should focus on addressing multiple pathophysiological targets, including insulin resistance, inflammation, oxidative stress, and fibrogenic pathways, to mitigate disease progression and improve long-term outcomes. This multifaceted approach will not only enhance patient care but also optimize the management of MASLD and its associated comorbidities [123].

## 5. Conclusions

The disease known as MASLD has replaced the term NAFLD, better reflecting the scientific evidence surrounding its pathophysiology and associated conditions. Thus, a deeper understanding of the molecular pathophysiology of MASLD and its diverse clinical manifestations is crucial for effective treatment and timely management (Figure 2).

The onset of MASLD is characterized by an excessive accumulation of lipids, particularly triglycerides, within hepatocytes. This accumulation results from an imbalance between lipid acquisition and disposal mechanisms. These metabolic disturbances lead to lipid overload, which, in turn, causes oxidative stress and mitochondrial dysfunction, setting the stage for hepatic inflammation. While lipid accumulation initiates MASLD, the progression to more severe forms, such as MASH, involves significant inflammatory processes. The presence of hepatic steatosis sensitizes the liver to various insults, leading to the activation of inflammatory pathways. Chronic inflammation then exacerbates liver injury, promoting fibrosis and increasing the risk of cirrhosis and hepatocellular carcinoma [114,181,182].

Although molecular research in MASLD has significantly advanced in recent years, there remains much to explore. A more comprehensive understanding will enable the development of innovative diagnostic and therapeutic strategies, ultimately having a substantial impact on clinical practice. One of the most active research areas is the pursuit of more precise and sensitive biomarkers to facilitate disease staging and predict progression.

These advancements are already beginning to make a significant impact on clinical practice and are expected to translate into earlier and more accurate diagnoses, as well as more personalized treatment strategies. A deeper understanding of the underlying molecular pathways will also enable the development of preventive strategies and better management of complications in advanced disease stages. In conclusion, the future of molecular research in MASLD is promising and is anticipated to have a critical impact on clinical practice.

## Figures and Tables

**Figure 1 ijms-26-02959-f001:**
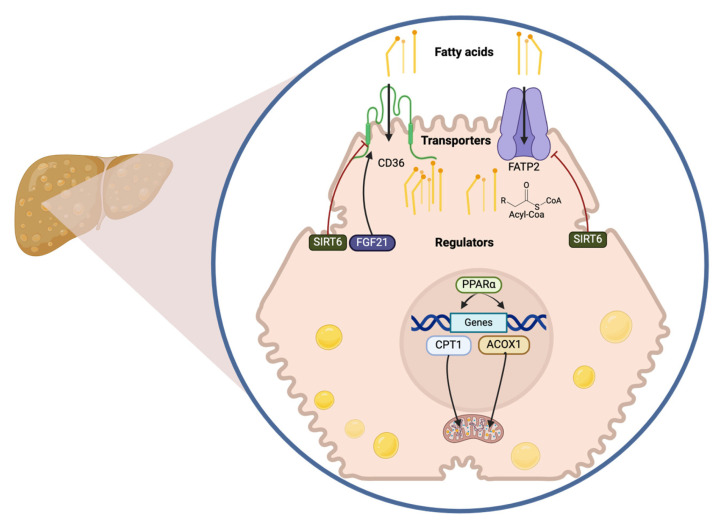
Fatty Acid Uptake in the Liver: Molecular Mechanisms in MASLD. Fatty acids are transported into the hepatocyte via CD36 and FATP2. SIRT6 influences fatty acid uptake by regulating *FGF21* and transporter expression. Within the nucleus, PPARα activates key lipid metabolism genes, including *CPT1* (involved in mitochondrial fatty acid oxidation) and *ACOX1* (associated with peroxisomal fatty acid oxidation). The mitochondria and lipid droplets are depicted as central components of lipid processing in hepatocytes.

**Figure 2 ijms-26-02959-f002:**
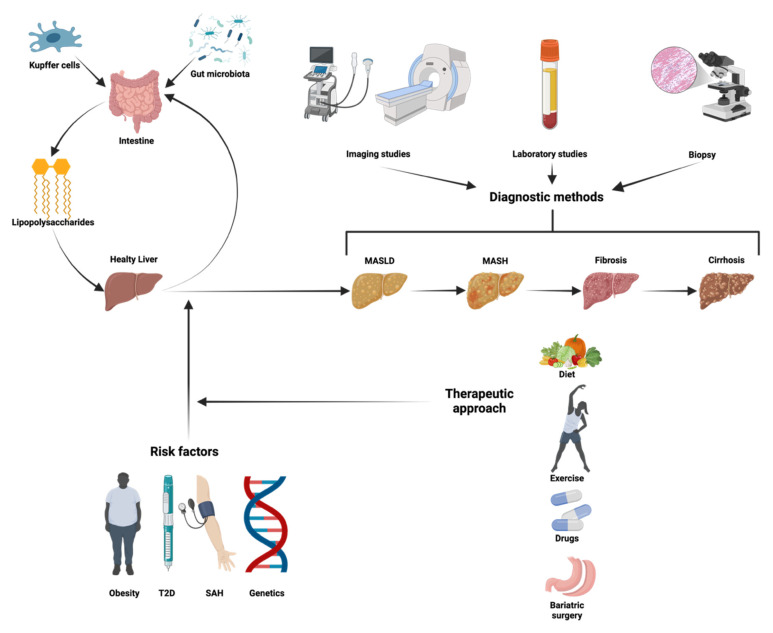
MASLD and associates. T2D; Type 2 diabetes, SAH; Systemic arterial hypertension, MASLD; Metabolic-dysfunction-associated steatotic liver disease, MASH; Metabolic-dysfunction-associated steatohepatitis.

**Table 1 ijms-26-02959-t001:** Molecular mechanisms involved in MASLD pathogenesis. ↑ increase, ↓ decrease.

Priority Level	Molecular Mechanism	Key Processes and Players	Relevance to MASLD Progression
1 (Core Driver)	Dysregulated Lipid Metabolism	Excessive fatty acid uptake (CD36, FATP2), enhanced de novo lipogenesis (SREBP1, FAS), impaired lipid export (MTTP, ApoB)	Central to hepatic steatosis; initiates lipid accumulation, leading to metabolic overload
2 (Core Driver)	Insulin Resistance and Hyperinsulinemia	Reduced AKT phosphorylation, increased SREBP1 activation, impaired glycogen synthesis	Amplifies lipogenesis and reduces lipid oxidation, promoting steatosis and metabolic stress
3 (Core Driver)	Mitochondrial Dysfunction and Oxidative Stress	Impaired β-oxidation (PPARα, ACSL5), reduced ATP production, increased ROS, mitophagy defects (BNIP3, p62)	Directly drives hepatocyte injury, apoptosis, and fibrogenesis
4 (Major Amplifier)	Chronic Inflammation	Kupffer cell activation, cytokine release (TNF-α, IL-6, CX3CL1), inflammasome activation (NLRP3)	Links metabolic dysfunction to liver fibrosis; perpetuates hepatocyte injury
5 (Major Amplifier)	Endoplasmic Reticulum (ER) Stress	ATF4-VLDLR axis, SIRT1 downregulation, unfolded protein response (UPR)	Exacerbates lipid accumulation and inflammatory signaling
6 (Major Amplifier)	Adipokine Dysregulation	↓ Adiponectin, FGF21; ↑ Leptin, Resistin, RBP4	Creates a pro-inflammatory, pro-fibrotic microenvironment in the liver
7 (Important Contributor)	Gut Microbiota Dysbiosis	↑ Intestinal permeability, ↑ LPS, ↑ TLR4/NF-κB signaling	Triggers hepatic inflammation via gut–liver axis
8 (Modifier of Susceptibility)	Genetic Factors	PNPLA3 I148M, TM6SF2, MBOAT7	Genetic variants modify lipid handling, inflammation, and fibrosis risk
9 (Modifier of Disease Course)	Epigenetic Modifications	miR-33 (↓ mitochondrial function), hypermethylation of ND6 (mitochondrial DNA)	Shapes disease severity through regulation of lipid metabolism, inflammation, and fibrosis

**Table 3 ijms-26-02959-t003:** Ongoing clinical trials.

Category	Treatment/Research	Mechanism/Target	Clinical Status/Research Goal
Thyroid Hormone Receptor Beta (THR-β) Agonists	Resmetirom (MGL-3196)	Liver-targeted THR-β agonist; reduces hepatic lipid content and fibrosis	Phase III (MAESTRO-NASH); evaluating efficacy in MASH patients with fibrosis
	VK2809	THR-β agonist; reduces liver fat content	Phase IIb; assessing efficacy in liver fat reduction
Fibroblast Growth Factor 21 (FGF21) Analogues	Efruxifermin (EFX)	FGF21 analogue; reverses liver fibrosis, improves insulin sensitivity	Phase IIb; showing fibrosis reversal, progressing to Phase III
Glucagon-Like Peptide-1 (GLP-1) Receptor Agonists	Semaglutide	GLP-1 receptor agonist; improves liver fibrosis, promotes weight loss	Phase III; demonstrated improvement in fibrosis, seeking regulatory approval
Dual GLP-1 and Glucagon Receptor Agonists	Survodutide	Dual GLP-1 and glucagon receptor agonist; targets liver fibrosis and metabolic dysfunction	Phase III; received FDA Breakthrough Therapy designation
Stearoyl-CoA Desaturase-1 (SCD1) Inhibitors	Denifanstat	SCD1 inhibitor; reduces hepatic lipid accumulation and inflammation	Phase III; evaluating in non-cirrhotic MASH patients

## Data Availability

No new data were created or generated in this manuscript. Data sharing is not applicable to this article.

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
