# Peer review of "Metabolic-Dysfunction-Associated Steatotic Liver Disease: Molecular Mechanisms, Clinical Implications, and Emerging Therapeutic Strategies"

_ijms, 2025, doi:10.3390/ijms26072959_

Round 1

Reviewer 1 Report

Comments and Suggestions for Authors

The article presents a comprehensive overview of metabolic dysfunction-associated steatotic liver disease (MASLD), covering its molecular mechanisms, clinical significance, and emerging therapeutic strategies. The author demonstrates a deep understanding of the subject, providing relevant arguments supported by logical reasoning and clear examples. One of the strengths of the article is its coherence and flow. The ideas are presented in a logical sequence, ensuring that readers can easily follow the argument.  However, several areas require further elaboration and clarification to enhance the depth, coherence, and impact of the review:

  • The transition from NAFLD to MASLD is a critical update in hepatology. However, the review does not provide sufficient discussion on the rationale and implications of this reclassification. The authors should elaborate on how this change affects clinical practice, diagnosis, and treatment paradigms.
  • While the review outlines multiple molecular mechanisms involved in MASLD pathogenesis, it would benefit from a clearer prioritization of these mechanisms. Are lipid metabolism alterations the primary driver, or does inflammation play a more central role? Including recent key studies or systematic reviews that establish these relationships would strengthen the discussion.
  • Given the rapid advancements in MASLD/MASH treatment, a section summarizing ongoing clinical trials and future research directions would further strengthen the review
  • Page 21 is blank
  • Page 22, 23: the table is not visible complete

Author Response

Reviewer 2

The article presents a comprehensive overview of metabolic dysfunction-associated steatotic liver disease (MASLD), covering its molecular mechanisms, clinical significance, and emerging therapeutic strategies. The author demonstrates a deep understanding of the subject, providing relevant arguments supported by logical reasoning and clear examples. One of the strengths of the article is its coherence and flow. The ideas are presented in a logical sequence, ensuring that readers can easily follow the argument.  However, several areas require further elaboration and clarification to enhance the depth, coherence, and impact of the review:

  • The transition from NAFLD to MASLD is a critical update in hepatology. However, the review does not provide sufficient discussion on the rationale and implications of this reclassification. The authors should elaborate on how this change affects clinical practice, diagnosis, and treatment paradigms.

A: Additional information has been included to complement the existing content. (Lines 91-100)

  • While the review outlines multiple molecular mechanisms involved in MASLD pathogenesis, it would benefit from a clearer prioritization of these mechanisms.

A: A table summarizing the prioritized molecular mechanisms involved in MASLD pathogenesis was added. (Table 1)

  • Are lipid metabolism alterations the primary driver, or does inflammation play a more central role? Including recent key studies or systematic reviews that establish these relationships would strengthen the discussion.

A: A paragraph has been added to the conclusion explaining how MASLD begins with lipid accumulation, and how the chronic progression of the disease ultimately leads to inflammation. Updated references have been included. (Lines 1101-1110)

  • Given the rapid advancements in MASLD/MASH treatment, a section summarizing ongoing clinical trials and future research directions would further strengthen the review

A: A table has been included to present the ongoing clinical trials (Table 3).

  • Page 21 is blank

A: The blank page on page 21 may be due to a defect in the PDF. We hope the current document displays correctly.

  • Page 22, 23: the table is not visible complete

A: The incomplete display of the table may be due to a defect in the PDF. We hope the current document is properly formatted and fully visible.

Reviewer 2 Report

Comments and Suggestions for Authors

Overall, this is a good literature review on an important topic of interest.

Some suggestions for improvement:

  1. Discuss MetALD and ALD in the introduction, and clearly state their differentiation from MASLD.
  2. From all these complex molecular mechanisms and mediators, please form an illustrative figure with the key mediators that play a role in MASLD and show their interactions. You don’t have to show all mediators, just the major ones, so that one can grasp the most important potential targets for therapeutic interventions.
  3. Please add a section for the role of adipokines in MASLD.
  4. The major ongoing phase 3 trial for GLP-1RAs and MASLD is the ESSENCE trial, which the authors should reference: doi: 10.1111/apt.18331
  5. When discussing MASLD, it is important to talk about its relevance to the recently proposed CRHM syndrome based on this publication: doi: 10.3390/biom15020213
  6. Please provide a clear diagnostic algorithm for the patients that need further evaluation. Specifically:
    • Which patients should be screened with LFTs and FIB-4?
    • Are there other scores better than FIB-4 (compare)?
    • Which patients need evaluation with abdominal ultrasound or elastography?
    • Is abdominal ultrasound useful by any means, or should all patients undergo elastography? (The 2024 guidelines of EASL do not mention ultrasound).
    • Which patients need biopsy?
    • Which patients need additional tests for differential diagnosis? For example, should all patients undergo hepatitis viral panels?

Author Response

Reviewer 1

Overall, this is a good literature review on an important topic of interest.

Some suggestions for improvement:

  1. Discuss MetALD and ALD in the introduction, and clearly state their differentiation from MASLD.

Answer (A): The information requested was added:

In recent years, a paradigm shift has redefined the classification of fatty liver diseases, grouping them under the broader term steatotic liver disease. This reclassification includes alcohol-associated liver disease (ALD), MASLD, and a newly recognized condition called metabolic dysfunction and alcohol-related liver disease (MetALD).

ALD refers to liver injury caused by chronic excessive alcohol consumption, with thresholds established by the World Health Organization and major liver societies. ALD is typically diagnosed when alcohol intake exceeds 21 standard drinks per week for men and 14 drinks per week for women. MetALD, on the other hand, is a newly introduced entity that identifies individuals with both metabolic dysfunction, such as obesity, type 2 diabetes, hypertension, or dyslipidemia, and moderate alcohol consumption, defined as intake levels below the ALD thresholds but above complete abstinence. This classification reflects growing evidence that metabolic risk factors, even in the presence of moderate alcohol intake, contribute synergistically to liver injury. By formally recognizing this overlap, MetALD provides a clearer diagnostic framework for a patient population that previously lacked precise classification within the traditional categories of alcohol-related or non-alcoholic liver disease. (Lines 62-77)

  1. From all these complex molecular mechanisms and mediators, please form an illustrative figure with the key mediators that play a role in MASLD and show their interactions. You don’t have to show all mediators, just the major ones, so that one can grasp the most important potential targets for therapeutic interventions.

A: A pertinent figures was made as suggested by the reviewer. (Figure )

  1. Please add a section for the role of adipokines in MASLD.

A: The section 2.6 Role of Adipokines in the Pathogenesis and Progression of MASLD/MASH has been added. (Lines 358-393)

  1. The major ongoing phase 3 trial for GLP-1RAs and MASLD is the ESSENCE trial, which the authors should reference: doi: 10.1111/apt.18331

A: The reference suggested by the reviewer was added. (Line 946)

  1. When discussing MASLD, it is important to talk about its relevance to the recently proposed CRHM syndrome based on this publication: doi: 10.3390/biom15020213

A: The reference suggested by the reviewer was added. (Line 147)

  1. Please provide a clear diagnostic algorithm for the patients that need further evaluation. Specifically:
    • Which patients should be screened with LFTs and FIB-4?
    • Are there other scores better than FIB-4 (compare)?
    • Which patients need evaluation with abdominal ultrasound or elastography?
    • Is abdominal ultrasound useful by any means, or should all patients undergo elastography? (The 2024 guidelines of EASL do not mention ultrasound).
    • Which patients need biopsy?
    • Which patients need additional tests for differential diagnosis? For example, should all patients undergo hepatitis viral panels?

A: All the points were addressed in the 3.3 Screening and Diagnostic Approach for MASLD section. (Lines 679-703)

Round 2

Reviewer 2 Report

Comments and Suggestions for Authors

Excellent work! The revised manuscript has been substantially improved. Figure 2 offers great mechanistic insights. I strongly suggest the manuscript for publication in its current form.

Author Response

I would like to inform you that Section 2.11, titled Unraveling the Spleen-Liver Axis, has been added to the document. (Lines 570-590) This section incorporates the references you suggested, ensuring a comprehensive and well-supported discussion of the topic.